# FLoRA-NA: Nearly Accurate Aggregation for Federated Low-Rank Adaptation

## Abstract

With the rapid emergence of foundation models and the increasing need for fine-tuning across distributed environments, Federated Low-Rank Adaptation (Fed-LoRA) has recently gained significant attention. Despite enormous potential, current FedLoRA methods face notable challenges due to inexact updates. Existing approaches have attempted to mitigate this issue, but they often introduce a *local-global generalization gap* and incur *substantial communication overhead*, limiting their scalability and effectiveness. To address these limitations, we propose **F**ederated **Lo**w-**R**ank **A**ggregation with **N**early **A**ccurate Estimation (FLoRA-NA). FLoRA-NA leverages the local LoRA matrices on the server to estimate the aggregated matrices $\hat{A}$ and $\hat{B}$, which are then distributed to clients for local updates. This surrogated aggregated matrices minimizes the divergence between ideal $\nabla \bar{W} = \sum_{u=1}^{U} B_u A_u$ and practical updates $\nabla \hat{W} = \hat{B}\hat{A}$ without adding communication cost beyond vanilla FedLoRA. By doing so, FLoRA-NA achieves communication efficiency and bridges the gap between local personalization and global generalization, addressing a key limitation of prior personalized FedLoRA approaches. We conduct extensive evaluations across diverse tasks, including natural language understanding, mathematical reasoning, and code-solving ability using various foundation models. Experimental results consistently demonstrate that FLoRA-NA achieves state-of-the-art global performance while maintaining low communication overhead.

## 1 Introduction

Current research increasingly focuses on developing pretrained Large Language Models (LLMs) that capture broad, cross-domain knowledge (Yang et al., 2025; Achiam et al., 2023; Touvron et al., 2023; Brown et al., 2020). However, this progress is increasingly constrained by the exhaustion of publicly available training data. To continue advancing these foundational models, researchers are turning to private or sensitive data sources, such as proprietary business data or user interactions on personal devices. However, this data is typically distributed across multiple parties, each possessing only a small amount that is insufficient for independently fine-tuning large models. Moreover, these parties are often restricted from sharing their data directly with others due to privacy or regulatory constraints. Federated Learning (FL) (Nguyen et al., 2025a; Tran et al., 2025; Huang et al., 2024; Nguyen et al., 2025b) offers a compelling framework to address this challenge by enabling collaborative, decentralized training across multiple devices without exchanging raw data.

Despite its potential, fine-tuning LLMs in federated settings poses notable difficulties due to the substantial computational and storage burdens placed on local clients, as well as the significant communication overhead required to synchronize large models. As a remedy, Low-Rank Adaptation (LoRA) (Hu et al., 2022) has been widely adopted. LoRA enables freezing the large pre-trained model weights and instead updating two small, injected low-rank matrices. Compared to full model fine-tuning, this approach typically achieves comparable, and sometimes even on-par, performance while substantially reducing computational costs and adding no inference latency. Motivated by these advantages, this work investigates LoRA-based strategies for federated fine-tuning of LLMs.

However, directly adapting LoRA methods from centralized settings and simply integrating them with FedAvg often results in **aggregation errors** (Sun et al., 2024; Guo et al., 2025), leading to suboptimal performance. Recent studies have attempted to mitigate these errors through various

strategies, including: (i) sharing only one LoRA matrix while keeping the other local to each client (Yi et al., 2023; Guo et al., 2025; Sun et al., 2024), (ii) transmitting additional information such as residual terms (Singhal et al., 2025), and (iii) sharing the complete set of stacked LoRA matrices from all participating clients (Wang et al., 2024). While these approaches have demonstrated certain improvements, they still face significant limitations that hinder the practicality of federated LoRA (FedLoRA) in real-world deployments. In this work, we conduct both theoretical and empirical analyses (see Section 2) and reveal two key findings. Firstly, sharing only one LoRA matrix enhances personalization but significantly diminishes the **global generalization capability** of the foundation model. Secondly, transmitting residual terms or stacked LoRA matrices introduces **substantial communication overhead**, which becomes increasingly problematic as the number of clients scales. Motivated by these findings, we aim to address the following central research question in this paper:

> *How can the complete set of low-rank matrices be effectively utilized to enhance the*
> *generalization performance of LLMs, while guaranteeing no information loss and incurring*
> *no additional communication overhead for the clients?*

To address the aforementioned challenges, we propose adaptive aggregation methods tailored for LoRA fine-tuning in federated settings: **F**ederated **Lo**w-**R**ank **A**ggregation with **N**early **A**ccurate Estimation (FLoRA-NA). FLoRA-NA aims to construct surrogate aggregated low-rank matrices $\hat{A}$ and $\hat{B}$ that more effectively approximate the aggregation $\nabla \bar{W} = \sum_{u \in \mathcal{U}} B_u A_u$ than the vanilla average of LoRA matrices $\bar{A}$ and $\bar{B}$. To demonstrate the flexibility and generalizability of our approach, we extend the FLoRA-NA to support other LoRA variants, including HiRA (Huang et al., 2025) and DoRA (Liu et al., 2024). We also conduct comprehensive experiments to demonstrate the superiority of FLoRA-NA over existing FedLoRA methods. Our evaluation covers both IID and non-IID NLP datasets with varying degrees of data heterogeneity.

## 2 RETHINKING FEDLORA

### 2.1 PRELIMINARIES: LOW-RANK ADAPTATION

LoRA is proposed based on the hypothesis that the change in weights during model adaptation has a low intrinsic rank, which is inspired by (Aghajanyan et al., 2021). Specifically, let $W \in \mathbb{R}^{k \times d}$ denote the pre-trained weight matrix, typically originating from a large-scale model. LoRA constrains its adaptation by introducing a low-rank decomposition of the update $W + \nabla W = W + BA$, where $B \in \mathbb{R}^{k \times r}$ and $A \in \mathbb{R}^{r \times d}$. The original weights $W$ are kept frozen during training, and only $A$ and $B$ are optimized. By selecting $r$ such that $r \ll \min(k, d)$, the number of trainable parameters is reduced significantly by an order of $O(r / \min(d, k))$ compared to full fine-tune. At initialization, $A$ is sampled from a Kaiming distribution, while $B$ is set to zero, ensuring $\nabla W = BA = 0$ at the start of training. This guarantees that the model's output initially matches that of the original pre-trained network, providing a stable starting point for fine-tuning. The update is scaled by $\alpha/r$, where $\alpha$ acts like a learning rate. This keeps the update magnitude consistent when changing $r$, enabling stable adaptation across different ranks.

### 2.2 FEDERATED LOW-RANK ADAPTATION

**Full-parameter matrix aggregation.** The early version (Qin et al., 2024) is full-parameter matrix aggregation. Suppose there are $U$ clients, each client $u$ possesses a private dataset $\mathcal{D}_u$. At round $i$, the client $u$ updates the local low-rank matrices as follows:

$$A_u^{(i)}, B_u^{(i)} = \underset{A_u^{(i-1)}, B_u^{(i-1)}}{\arg\min} \mathcal{L}(\mathcal{D}_u; W_0 + \nabla W_u^{(i-1)}), \text{ s.t. } \nabla W_u^{(i-1)} = B_u^{(i-1)} A_u^{(i-1)}. \quad (1)$$

After the local parameters are received, the server updates the full-parameter aggregated matrix and sends back to local clients the learned gradient $\nabla W^{(i)} = \frac{1}{U} \sum_{u=1}^{U} B_u^{(i)} A_u^{(i)}$. However, it introduces the communication overhead when fine-tuning LLMs with an extremely large number of parameters.

**Separate matrix aggregation.** Separate matrix aggregation is a framework that combines the principles of FL and LoRA for fine-tuning LLMs. According to (Xie et al., 2023), at the beginning of training, all clients are initialized with the same LLM. Instead of fully fine-tuning the LLM, each

client adopts LoRA to update the model efficiently. Unlike the full-rank matrix aggregation, where the full matrix $\nabla W^{(i)} = \frac{1}{U} \sum_{u=1}^{U} B_u^{(i)} A_u^{(i)}$ is broadcasted to the local clients. At each communication round $r$, the server broadcasts the aggregated matrices $\bar{A}^{(i)}, \bar{B}^{(i)}$ such that:

$$\bar{A}^{(i)} = \frac{1}{U} \sum_{u=1}^{U} A_u^{(i)} \text{ and } \bar{B}^{(i)} = \frac{1}{U} \sum_{u=1}^{U} B_u^{(i)}. \tag{2}$$

The separate matrix aggregation induces a challenge, where the separate matrix aggregation leads to a difference with the joint matrix aggregation, i.e.,

$$\bar{B}^{(i)} \bar{A}^{(i)} = \left( \frac{1}{U} \sum_{u=1}^{U} B_u^{(i)} \right) \left( \frac{1}{U} \sum_{u=1}^{U} A_u^{(i)} \right) \neq \frac{1}{U} \sum_{u=1}^{U} (B_u^{(i)} A_u^{(i)}). \tag{3}$$

The discrepancy between these two formulations introduces an aggregation error, potentially degrading the performance of the global model. In real-world FL settings, client datasets are often non-identically distributed (non-IID). This heterogeneity induces domain shifts, making it challenging for the aggregated global model to generalize well across diverse client distributions. Addressing these challenges is crucial for achieving both accuracy and robustness in FedLoRA.

**Single matrix aggregation.** This approach focuses on the strategy of freezing certain model components while locally training a specific matrix to enhance personalization. Only one matrix is aggregated across clients, while the remaining matrices are kept fixed at the server, effectively acting as frozen parameters (Sun et al., 2024) or being trained locally (Guo et al., 2025). This design has demonstrated strong personalized performance. However, such methods often fail to adequately bridge the gap between personalization and generalization.

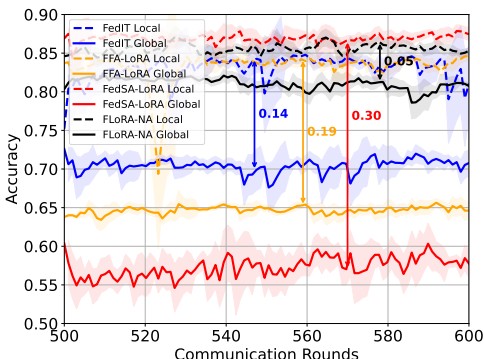

Figure 1: We evaluate leading FedLoRA methods on MNLI dataset and observe a notable gap between local and global test accuracy. Our proposed method FLoRA-NA shows state-of-art robustness by mitigating inter-client divergence throughout the learning process, leading to a reduced local-global generalization gap.

To investigate this limitation, we conduct experiments and present the results in Fig. 1, where the local models and global models are evaluated on local and global test datasets, respectively. The details of evaluation metric is detailed in Appendix C.4. As shown, the generalization gap, measured as the difference between local and global accuracy, is substantially large for methods employing single matrix aggregation. This indicates that while the local personalized models adapt well to their respective datasets, they tend to overfit to local data and fail to leverage valuable knowledge from other clients, ultimately compromising generalization.

**Stacked matrix aggregation.** In stacked matrix aggregation (Wang et al., 2024), the low-rank matrices from local users are stacked into a joint matrix $\bar{A}^{(i)} = \bigoplus_{u=1}^{U} A_u^{(i)}, \bar{B}^{(i)} = \bigoplus_{u=1}^{U} B_u^{(i)}$, where $A_u^{(i)} \in \mathbb{R}^{r_u \times d}, B_u^{(i)} \in \mathbb{R}^{k \times r_u}, \bar{A}^{(i)} \in \mathbb{R}^{(\sum_{u=1}^{U} r_u) \times d}, \bar{B}^{(i)} \in \mathbb{R}^{k \times (\sum_{u=1}^{U} r_u)}$. Instead of broadcasting to each user low-rank matrices with size $(r_u \times k) + (r_u \times d)$, the stacked matrix aggregation requires the communication cost of $((\sum_{u=1}^{U} r_u) \times k) + ((\sum_{u=1}^{U} r_u) \times d)$. As a consequence, stacked matrix scales the communication overhead with the number of clients $U$, thus, inducing significant communication overhead in FedLoRA system with a large number of users. (Wang et al., 2024) also propose directly adding $\bar{B}^{(i)} \bar{A}^{(i)}$ to the global and local pretrained weights, followed by reinitializing both $A$ and $B$ in every round, which induces significant noise and slows down training (see Section 5).

## 3 METHODOLOGY

To avoid incurring additional communication overhead while enhancing the generalization capability of deploying LLMs in federated systems, our approach leverages the client-specific matrices $B_u^{(i)}$

and $A_u^{(i)}, \forall u \in U$, which are already available at the aggregation server. The objective is to design an aggregation algorithm that produces global matrices $\bar{B}^{(i)}$ and $\bar{A}^{(i)}$ such that their product approximates the average of the local matrix products across all clients. For instance,

$$\bar{B}^{(i)} \bar{A}^{(i)} \approx \frac{1}{U} \sum_{u=1}^{U} \left( B_u^{(i)} A_u^{(i)} \right), \tag{4}$$

thereby preserving the generalization capability of vanilla FL. Furthermore, the aggregated matrices $\bar{B}^{(i)}$ and $\bar{A}^{(i)}$ must be of compatible dimensions with the local LoRA matrices $B_u^{(i)}$ and $A_u^{(i)}$ to ensure seamless integration with client models.

---

**Algorithm 1** Federated Low-Rank Adaptation via Nearly-accurate Aggregation.

---

**Input:** set of clients $\mathcal{U}$, number of communication rounds $R$, local learning rate $\eta$, global learning rate $\eta_g$, searching space hyper-parameter $\kappa$.
**Initialize:** Pretrained model $W$, LoRA matrices $\bar{A}^{(i)}, \bar{B}^{(i)}$.
1: **while** $i \leq R$ **do**                                                              ▷ global iterations
2:     Update $\bar{A}^{(i)}, \bar{B}^{(i)}$ to clients.
3:     **for** $1, \ldots, U$ **do**                                                       ▷ client update
4:         Solve $A_u^{(i,E)}, B_u^{(i,E)} = \arg\min_{A,B} \mathcal{E}(\theta_u^{(i,0)} + BA, \mathcal{D}_u)$ after $E$ epochs.
5:         Upload client's LoRA matrices to server: $A_u^{(i)} \leftarrow A_u^{(i,E)}, B_u^{(i)} \leftarrow B_u^{(i,E)}$.
6:     **end for**
7:     Solve for $P^* = \{P_1^*, \ldots, P_U^*\}$ and $Q^* = \{Q_1^*, \ldots, Q_U^*\}$ via (6).     ▷ Nearly accurate aggregation
8:     Measure $\bar{A}^{(i)} = \sum_{u=1}^{U} Q_u^* A_u^{(i)}$ and $\bar{B}^{(i)} = \sum_{u=1}^{U} P_u^* B_u^{(i)}$.
9: **end while**

---

To matrix product approximation in (4), one natural idea is to have the server that finding the optimal solution $\bar{B}^{(i)}$ and $\bar{A}^{(i)}$ such that $\bar{B}^{(i)}, \bar{A}^{(i)} = \arg\min_{\bar{B}, \bar{A}} \left[ \bar{B}\bar{A} - \frac{1}{U} \sum_{u=1}^{U} \left( B_u^{(i)} A_u^{(i)} \right) \right]$. However, it is noteworthy that: 1) LoRA matrices have a large number of parameters, which requires a significantly high computation overhead. 2) The data is limited, as only LoRA matrices from the available clients are leveraged as input data for the optimization, making learning $\bar{B}^{(i)}, \bar{A}^{(i)}$ is challenging, especially when the dimensionality of $\bar{B}^{(i)}, \bar{A}^{(i)}$ is larger than that of the data.

Acknowledging these two challenges, we turn to find the surrogate solution of vectors $P = \{P_u | u \in U\}, Q = \{Q_u | u \in U\}$, where $P, Q \in \mathbb{R}^{U \times 1}$. These vectors can be interpreted as transformation vectors that determine how each client's local LoRA gradients are linearly combined during aggregation. Specifically, $\bar{B}^{(i)} = \sum_{u=1}^{U} P_u B_u^{(i)}$, and $\bar{A}^{(i)} = \sum_{u=1}^{U} Q_u A_u^{(i)}$. As a consequence, the nearly accurate aggregation at the server is found via the following:

$$\bar{B}^{(i)} = \sum_{u=1}^{U} P_u^* B_u^{(i)}, \ \bar{A}^{(i)} = \sum_{u=1}^{U} Q_u^* A_u^{(i)}, \quad P^* = \{P_u^* | u \in U\}, \ Q^* = \{Q_u^* | u \in U\}, \tag{5}$$

where $P^*, Q^*$ are the optimal coefficients and learned via the following optimization problem:

$$P^*, Q^* = \arg\min_{P,Q} \left[ \sum_{u=1}^{U} P_u B_u^{(i)} \times \sum_{u=1}^{U} Q_u A_u^{(i)} - \frac{1}{U} \sum_{u=1}^{U} \left( B_u^{(i)} A_u^{(i)} \right) \right]. \tag{6}$$

The details of FLoRA-NA are presented in Algorithm 1. Furthermore, we summarize several key observations regarding the proposed FLoRA-NA as follows:

**Corollary 1** (Computation Efficiency). *Since $P, Q \in \mathbb{R}^{U \times 1}$ and $U \ll k \times d$, the optimization involving the two learnable coefficients $P$ and $Q$ is significantly simpler compared to directly minimizing over the LoRA matrices $A$ and $B$. Consequently, the proposed approach achieves near-accurate aggregation with substantially improved computational efficiency.*

**Corollary 2** (Communication Efficiency). *Since the nearly accurate aggregation performed on the server utilizes the local LoRA matrices $A_u^{(i)}$ and $B_u^{(i)}$ sent by the clients and returns the aggregated matrices $\bar{A}^{(i)}$ and $\bar{B}^{(i)}$ with the same dimensionality as the standard averaged LoRA matrices, it does not incur any additional communication cost compared to FedIT.*

**Discussion.** FLoRA-NA surpasses prior approaches by making the nearly exact aggregation without causing any further communication overhead. In practical, we can apply various compression techniques, e.g., sparsification (Liang et al., 2025), quantization (Gao et al., 2025; Dettmers et al., 2023; Xia et al., 2024), on the aggregated LoRA matrices to achieve further communication efficiency. We show corresponding experiments in Section 5.6.

## 4 CONVERGENCE ANALYSIS

Numerous studies have examined the convergence rate of FedLoRA (Chen et al., 2025; Guo et al., 2025) by treating the LoRA matrices $A$ and $B$ as separate entities. However, as discussed in Section 2, the key distinction between FedLoRA and vanilla FL lies in the *aggregation gap* arising from the difference between averaging the full-parameter model and separately aggregating the low-rank matrices. Consequently, in this work, we begin by formally defining the distance between these two aggregation approaches. This distance is then incorporated into the convergence proof of vanilla FL from (Jhunjhunwala et al., 2023). By doing so, we directly demonstrate how our method reduces this distance, thereby effectively bridging the gap between FedLoRA and vanilla FL.

**Definition 1.** *Let $\varrho$ denote the divergence between the ideal aggregated low-rank matrix and the aggregated matrix produced by any FedLoRA algorithm. Specifically, we define the ideal aggregated low-rank matrix as $\nabla \bar{W} = \frac{1}{U} \sum_{u=1}^{U} B_u A_u$ and the FedLoRA-aggregated matrix as $\nabla \hat{W} = \hat{B} \hat{A}$. The divergence is then given by $\varrho = \|\nabla \bar{W} - \nabla \hat{W}\|^2$.*

We are now in a position to analyze the convergence behavior of the global model. The general convergence theorem for FLoRA-NA is given as follows:

**Theorem 1.** *[Federated LoRA convergence rate, $\mathcal{L}_u$ are convex] Under Assumptions 2 and 3, and assuming clients compute full-batch gradients with full participation and $\eta \leq \frac{1}{6EL}$, the iterates $\{W^{(i)}\}$ satisfy,*

$$\mathcal{L}(W^{(i)}) - \mathcal{L}(W^*) \leq \mathcal{O}\left(\frac{\|W^{(0)} - W^*\|^2}{\sum_{i=0}^{R-1} \eta E}\right) + \mathcal{O}\left(\eta E \sigma_*^2\right) + \mathcal{O}\left(\eta^2 E(E-1)L\sigma_*^2\right) + \mathcal{O}\left(kd\varrho\right).$$

*where $\varrho$ is the divergence between the FedLoRA aggregated matrices $\nabla \hat{W}$ and the ideal aggregated low-rank matrix $\nabla \bar{W} = \frac{1}{U} \sum_{u=1}^{U} B_u A_u$.*

According to Theorem 1, the current federated LoRA frameworks increase the bounds of optimum convergence by a value that is proportional to $\mathcal{O}\left(kd\varrho\right)$ under smooth and non-convex conditions. In FLoRA-NA, our contribution is to minimize the divergence $\mathcal{O}\left(kd\varrho\right)$. As a consequence, we can achieve the better convergence bounds. For instance,

**Corollary 3.** *[FLoRA-NA convergence rate, $\mathcal{L}_u$ are convex] Under Assumptions 2 and 3 and assuming clients compute full-batch gradients with full participation and $\eta \leq \frac{1}{6EL}$, the iterates $\{W^{(i)}\}$ generated by FLoRA-NA satisfy,*

$$\mathcal{L}(W^{(i)}) - \mathcal{L}(W^*) \leq \mathcal{O}\left(\frac{\|W^{(0)} - W^*\|^2}{\sum_{i=0}^{R-1} \eta E}\right) + \mathcal{O}\left(\eta E \sigma_*^2\right) + \mathcal{O}\left(\eta^2 E(E-1)L\sigma_*^2\right) + \mathcal{O}\left(kd\varrho_{\textit{FLoRA-NA}}\right).$$

*where $\varrho_{\textit{FLoRA-NA}}$ is the divergence between the FLoRA-NA aggregated matrices $\nabla \hat{W}$ and the ideal aggregated low-rank matrix $\nabla \bar{W} = \frac{1}{U} \sum_{u=1}^{U} B_u A_u$, and $\varrho_{\textit{FLoRA-NA}} \leq \epsilon \ll \varrho$.*

Here, we have $\varrho_{\text{FLoRA-NA}} \leq \epsilon \ll \varrho$ due to the optimization in equation 6.

**Non-convexity.** In this case, we need the data heterogeneity to be bounded everywhere as follows.

**Assumption 1.** *There exists a constant $\sigma_g^2 > 0$ such that the global gradient variance is bounded as follows. $\frac{1}{U} \sum_{u=1}^{U} \|\nabla \mathcal{L}(\mathcal{D}_u; W) - \mathcal{L}(\mathcal{D}; W)\| \leq \sigma^2, \forall W \in \mathbb{R}^d$.*

**Theorem 2.** *[Federated LoRA convergence rate, $\mathcal{L}_u$ are non-convex] Under Assumptions 2,3 and assuming clients compute full-batch gradients with full participation and $\eta \leq \frac{1}{6EL}$, the iterates $\{W^{(i)}\}$ satisfy,*

$$\min_{i \in [R]} \left\|B^{(i)} A^{(i)}\right\|^2 \leq \mathcal{O}\left(\frac{(\mathcal{L}(W^{(0)}) - \mathcal{L}^*)}{\sum_{i=0}^{R-1} \eta E}\right) + \mathcal{O}\left(\eta^2 L^2 E(E-1)\sigma_g^2\right) + \mathcal{O}\left(\eta L E \sigma_g^2\right) + \mathcal{O}\left(kd\varrho\right).$$

**Corollary 4.** *[FLoRA-NA convergence rate, $\mathcal{L}_u$ are convex] Under Assumptions 2,3 and assuming clients compute full-batch gradients with full participation and $\eta \leq \frac{1}{6EL}$, the iterates $\{W^{(i)}\}$ generated by FLoRA-NA satisfy,*

$$\min_{i \in [R]} \left\| B^{(i)} A^{(i)} \right\|^2 \leq \mathcal{O}\left(\frac{(\mathcal{L}(W^{(0)}) - \mathcal{L}^*)}{\sum_{i=0}^{R-1} \eta E}\right) + \mathcal{O}\left(\eta^2 L^2 E(E-1)\sigma_g^2\right) + \mathcal{O}\left(\eta L E \sigma_g^2\right)$$
$$+ \mathcal{O}\left(k d \varrho_{FLoRA\text{-}NA}\right).$$

*where $\varrho_{FLoRA\text{-}NA}$ is the divergence between the FLoRA-NA aggregated matrices $\nabla \hat{W}$ and the ideal aggregated low-rank matrix $\nabla \bar{W} = \frac{1}{U} \sum_{u=1}^{U} B_u A_u$, $\varrho_{FLoRA\text{-}NA} \leq \epsilon \ll \varrho$.*

**Discussion.** In Vanilla FL, the error can be bounded by $\mathcal{O}\left(\frac{(\mathcal{L}(W^{(0)}) - \mathcal{L}^*)}{\sum_{i=0}^{R-1} \eta E}\right) + \mathcal{O}\left(\eta^2 L^2 E(E-1)\sigma_g^2\right)$ in convex settings, and by $\mathcal{O}\left(\frac{(\mathcal{L}(W^{(0)}) - \mathcal{L}^*)}{\sum_{i=0}^{R-1} \eta E}\right) + \mathcal{O}\left(\eta^2 L^2 E(E-1)\sigma_g^2\right)$ in non-convex settings. The key difference between FedLoRA and FedAvg comes from the divergence $\mathcal{O}\left(k d \varrho\right)$ between the full rank gradients and the low-rank gradients composed of two aggregated matrices $\hat{A}, \hat{B}$. The difference between the federated LoRA algorithms also leads to the difference in the divergence value $\varrho$, and thus, making a difference in $\mathcal{O}\left(k d \varrho\right)$. FLoRA-NA directly minimizes the divergence (i.e., $\varrho_{\text{FLoRA-NA}} = \min \varrho$), and thus, directly improves the generalization of the federated LoRA system.

## 5 EXPERIMENTAL EVALUATIONS

Our experiments are built on the FederatedScope-LLM (Kuang et al., 2024). We first conduct empirical studies to demonstrate the effectiveness of our nearly accurate mechanism. We then evaluate FLoRA-NA effectiveness through a series of experiments designed to answer three key questions: (1) Does FLoRA-NA deliver a global model with strong cross-client generalization? (2) Does FLoRA-NA boost robustness under varying client numbers and data heterogeneity? (3) Does FLoRA-NA fit the practical scenarios in terms of communication and computation? All results use five random seeds to ensure statistical reliability. Implementation details are detailed in Appendix C.

### 5.1 NEARLY ACCURATE ESTIMATION EFFICIENCY

To evaluate the efficiency of the nearly accurate mechanism, we compute the *normalized Frobenius norm of divergence* between the learned aggregated model $\bar{W} = \bar{B}^{(i)} \bar{A}^{(i)}$, and the ideal aggregated model $\bar{W} = \sum_{u=1}^{U} B_u^{(i)} A_u^{(i)}$. Figure 2 shows the effectiveness of the proposed nearly accurate

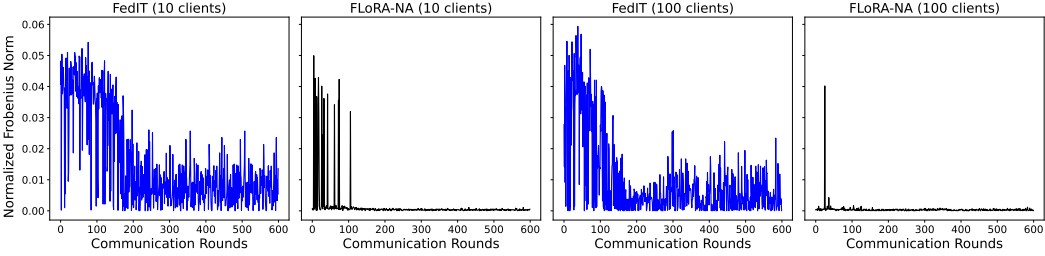

Figure 2: Comparison of the normalized Frobenius norm of divergence between the gradient obtained from the ideal update and that from the approximate update under the naive FedAvg strategy with full-parameter, using FedIT and the proposed FLoRA-NA method on MNLI dataset.

mechanism in various FL system sizes. Specifically, FedIT suffers from persistent and highly fluctuating aggregation loss across both small and large-scale client settings. Meanwhile, FLoRA-NA provides a significantly more accurate approximation that is very close to the ideal gradient update, ensuring robustness and scalability even when the number of participating clients scales up. We further show the layer-wise normalized Frobenius norm of divergence at the beginning and at the end of nearly accurate optimization at communication round 50 in Appendix D.2

## 5.2 Is nearly accurate better than SVD-based aggregation?

Many approaches have explored the use of SVD (Singhal et al., 2025; Ning et al., 2025) to decompose the gradient update matrix, $\nabla W$, into the LoRA matrices $A$ and $B$. To assess the robustness of FLoRA-NA, we implement SVD-based variants applied to $\nabla W$ and measure the execution time required to compute the corresponding matrices. As shown in Table 1, both standard SVD and Gram-Schmidt-based SVD incur significantly higher computational costs, with execution time approximately 10 to 15 times greater than that of FLoRA-NA. This substantial overhead arises from the computational complexity of SVD, which scales as $\mathcal{O}(kd^2)$. Moreover, SVD-based methods may introduce numerical inaccuracies due to floating-point precision errors, potentially degrading the overall performance of FedLoRA.

Table 1: Per-round execution time and normalized Frobenius norm of divergence of FLoRA-NA compared to other weight decomposition methods on the MNLI dataset using the RoBERTa-large model. The training settings are the same as in Appendix C.

| Method (32 bits) | Execution time | Frobenius Norm |
|---|---|---|
| Gram-Schmidt($W_q, W_v$) | 7.99s | $0.04 \pm 0.008$ |
| SVD ($W_q, W_v$) | 19.77s | $0.03 \pm 0.006$ |
| Gram-Schmidt($W_q, W_k, W_v, W_o$) | 15.98s | $0.06 \pm 0.013$ |
| SVD ($W_q, W_k, W_v, W_o$) | 39.54s | $0.06 \pm 0.011$ |
| FLoRA-NA (100 steps) | 1.32s | $0.003 \pm 0.0007$ |
| Method (16 bits) | Execution time | Frobenius Norm |
| Gram-Schmidt($W_q, W_v$) | 3.81s | $0.10 \pm 0.025$ |
| SVD ($W_q, W_v$) | 8.45s | $0.06 \pm 0.017$ |
| Gram-Schmidt($W_q, W_k, W_v, W_o$) | 7.18s | $0.11 \pm 0.027$ |
| SVD ($W_q, W_k, W_v, W_o$) | 16.77s | $0.08 \pm 0.02$ |
| FLoRA-NA (100 steps) | 0.52s | $0.005 \pm 0.0012$ |

## 5.3 Global model generalization

Table 2: Performance comparison on the GLUE benchmark datasets across different LoRA variants. Each result is reported in the format $\{x \mid y\}$, where $x$ denotes the local evaluation performance and $y$ denotes the global evaluation performance. The best result in each sub-column is underlined.

| Variant | Method | MNLI | SST-2 | MRPC | QNLI | QQP | RTE | STS-B | Avg |
|---|---|---|---|---|---|---|---|---|---|
| LoRA (ICLR'22) | FedIT-LoRA | 84.41 \| 71.54 | 91.53 \| 79.56 | 83.90 \| 68.23 | 86.44 \| 72.84 | 82.75 \| 68.10 | 81.01 \| 65.18 | 86.63 \| 73.15 | 85.24 \| 71.23 |
| | FedDPA-LoRA | 86.51 \| 69.22 | 93.87 \| 78.27 | 85.85 \| 66.84 | 89.34 \| 74.71 | 85.28 \| 69.73 | 82.81 \| 63.32 | 88.11 \| 72.29 | 87.40 \| 70.02 |
| | FFA-LoRA | 84.70 \| 65.25 | 90.91 \| 75.86 | 84.01 \| 61.45 | 86.27 \| 69.80 | 83.02 \| 64.39 | 80.62 \| 61.57 | 86.31 \| 69.90 | 85.12 \| 66.89 |
| | FedSA-LoRA | 87.08 \| 58.33 | 94.09 \| 72.21 | 85.13 \| 58.40 | 90.86 \| 63.47 | 85.98 \| 61.92 | 82.44 \| 55.08 | 88.27 \| 65.40 | 87.69 \| 62.12 |
| | FLoRA | 85.73 \| 74.11 | 91.86 \| 83.05 | 82.30 \| 73.91 | 87.07 \| 78.51 | 82.17 \| 75.93 | 80.24 \| 69.41 | 87.63 \| 73.15 | 85.29 \| 75.44 |
| | FedEx-LoRA | 85.72 \| 77.05 | 90.48 \| 84.92 | 83.10 \| 77.67 | 87.36 \| 80.31 | 82.01 \| 76.24 | 79.41 \| 73.85 | 86.12 \| 80.06 | 84.89 \| 78.58 |
| | FLoRA-NA | 85.63 \| 81.13 | 91.94 \| 87.05 | 83.78 \| 79.81 | 88.19 \| 84.59 | 84.21 \| 79.77 | 81.94 \| 76.43 | 88.04 \| 83.86 | 86.25 \| 81.81 |
| DoRA (ICML'24) | FedIT-DoRA | 86.72 \| 75.11 | 93.84 \| 83.02 | 85.12 \| 70.77 | 89.65 \| 73.33 | 85.01 \| 70.41 | 83.43 \| 67.80 | 88.75 \| 74.41 | 87.50 \| 73.55 |
| | FedDPA-DoRA | 88.63 \| 72.07 | 94.92 \| 79.22 | 86.03 \| 71.01 | 90.24 \| 71.12 | 87.44 \| 70.93 | 84.92 \| 65.48 | 89.02 \| 74.82 | 88.74 \| 72.09 |
| | FFA-DoRA | 87.93 \| 67.70 | 93.05 \| 77.91 | 85.20 \| 64.02 | 87.39 \| 70.97 | 85.15 \| 68.38 | 81.92 \| 64.44 | 89.54 \| 74.43 | 87.17 \| 69.69 |
| | FedSA-DoRA | 88.25 \| 62.20 | 95.13 \| 75.01 | 85.31 \| 59.14 | 91.92 \| 68.03 | 88.05 \| 63.87 | 84.12 \| 57.02 | 89.34 \| 68.92 | 88.87 \| 64.88 |
| | FDoRA | 86.90 \| 78.16 | 92.43 \| 83.44 | 85.73 \| 78.24 | 88.61 \| 82.73 | 85.22 \| 77.90 | 82.28 \| 73.48 | 88.17 \| 81.41 | 87.05 \| 79.34 |
| | FedEx-DoRA | 87.73 \| 80.44 | 93.14 \| 86.76 | 85.59 \| 78.66 | 89.10 \| 83.14 | 85.94 \| 77.58 | 83.61 \| 75.02 | 88.97 \| 83.12 | 87.73 \| 80.67 |
| | FDoRA-NA | 86.92 \| 83.54 | 92.97 \| 88.17 | 85.02 \| 80.04 | 90.34 \| 86.82 | 85.37 \| 82.01 | 84.22 \| 77.74 | 90.14 \| 85.97 | 87.85 \| 83.47 |
| HiRA (ICLR'25) | FedIT-HiRA | 87.11 \| 75.59 | 94.24 \| 82.40 | 86.78 \| 71.22 | 89.12 \| 74.71 | 85.65 \| 70.92 | 84.14 \| 67.20 | 88.21 \| 74.93 | 87.89 \| 73.85 |
| | FedDPA-HiRA | 89.01 \| 74.79 | 95.24 \| 80.67 | 88.51 \| 67.65 | 91.81 \| 74.71 | 87.92 \| 71.48 | 84.35 \| 66.92 | 89.54 \| 72.33 | 89.48 \| 72.65 |
| | FFA-HiRA | 86.25 \| 69.33 | 92.48 \| 78.20 | 85.91 \| 65.63 | 87.91 \| 72.41 | 84.72 \| 67.91 | 82.31 \| 63.77 | 87.91 \| 72.83 | 86.78 \| 70.01 |
| | FedSA-HiRA | 88.62 \| 60.70 | 95.41 \| 74.44 | 87.74 \| 60.71 | 93.33 \| 68.39 | 88.59 \| 64.44 | 84.74 \| 57.51 | 90.71 \| 69.38 | 89.88 \| 65.08 |
| | FHiRA | 87.47 \| 82.37 | 92.21 \| 83.15 | 86.66 \| 77.67 | 86.27 \| 80.18 | 85.90 \| 77.82 | 83.65 \| 73.11 | 88.74 \| 80.02 | 87.27 \| 79.19 |
| | FedEx-HiRA | 87.94 \| 83.92 | 93.13 \| 85.57 | 86.84 \| 78.76 | 87.74 \| 82.62 | 86.49 \| 79.65 | 83.07 \| 74.39 | 89.08 \| 83.71 | 87.76 \| 81.05 |
| | FHiRA-NA | 88.41 \| 85.03 | 94.42 \| 88.61 | 87.53 \| 80.62 | 90.74 \| 86.21 | 86.88 \| 82.53 | 84.65 \| 78.11 | 89.54 \| 86.33 | 88.87 \| 83.92 |

**Results on Natural Language Understanding Tasks.** As shown in Table 2, FLoRA-NA consistently achieves the state-of-the-art generalization performance when paired with various LoRA variants. In contrast, FedSA-LoRA (and occasionally FedDPA-LoRA) often obtains the strongest results on local evaluations, but its performance drops significantly on global evaluations, highlighting the overfitting effect caused by personalization. Meanwhile, FedEx-LoRA attains accuracy close to FLoRA-NA. However, it remains lower since the residual error is added to the frozen pretrained model, which ensures the accuracy of forward update, but cannot fully participate in gradient updates of $A$ and $B$ matrices.

**Results on Mathematical Reasoning and Code-solving Tasks.** As shown in Table 3, FLoRA-NA consistently demonstrates strong generalization on both mathematical reasoning and code-generation tasks, while also scaling effectively with increasingly powerful backbones. We further show a generated example for mathematical reasoning in Appendix D.3 and code-solving in Appendix D.4.

## 5.4 Robustness under varying data heterogeneity and client numbers

**Data heterogeneity.** Figure 3 llustrates the test accuracy on global model across varying levels of data heterogeneity for MNLI, SST-2, MRPC and RTE datasets. As shown in the figure, all

Table 3: Performance comparison on mathematical reasoning tasks (GSM8K and MATH) (Cobbe et al., 2021) and code-solving (HumanEval and MBPP) tasks using various backbones. Each result is reported in the format $\{x \mid y\}$, where $x$ denotes the local evaluation performance and $y$ denotes the global evaluation performance. The best result in each sub-column of each variant is underlined.

| Model | Method | GSM8K | | MATH | | HumanEval | | MBPP | | Avg | |
|---|---|---|---|---|---|---|---|---|---|---|---|
| LLaMA-2-7B | FedIT | 46.23 | 35.68 | 6.51 | 4.56 | 21.32 | 15.74 | 35.10 | 28.37 | 27.29 | 21.09 |
| | FFA-LoRA | 46.32 | 33.12 | 6.63 | 4.28 | 21.45 | 14.92 | 34.80 | 27.15 | 27.30 | 19.87 |
| | FedSA-LoRA | 46.63 | 28.82 | 7.13 | 3.57 | 22.01 | 13.40 | 36.25 | 25.02 | 28.01 | 17.70 |
| | FLoRA | 46.27 | 38.71 | 6.42 | 4.98 | 21.22 | 15.88 | 35.67 | 28.53 | 27.39 | 22.03 |
| | FedEx-LoRA | 46.37 | 39.19 | 6.37 | 5.22 | 21.18 | 16.30 | 35.51 | 30.95 | 27.36 | 22.91 |
| | FLoRA-NA | 46.41 | 42.89 | 6.87 | 5.86 | 21.78 | 18.93 | 35.92 | 32.10 | 27.75 | 24.95 |
| Mistral-7B | FedIT | 66.54 | 51.59 | 19.89 | 12.81 | 44.45 | 36.92 | 58.10 | 49.81 | 47.25 | 37.78 |
| | FFA-LoRA | 67.32 | 50.18 | 19.70 | 12.19 | 44.72 | 35.85 | 57.95 | 48.42 | 47.42 | 36.66 |
| | FedSA-LoRA | 69.93 | 43.20 | 21.34 | 8.32 | 46.10 | 31.13 | 61.20 | 43.70 | 49.64 | 31.49 |
| | FLoRA | 67.44 | 59.32 | 20.01 | 13.53 | 45.29 | 37.94 | 59.18 | 51.16 | 47.98 | 40.49 |
| | FedEx-LoRA | 67.12 | 60.40 | 19.92 | 14.17 | 44.67 | 39.15 | 59.66 | 53.18 | 47.84 | 41.73 |
| | FLoRA-NA | 67.44 | 63.40 | 20.37 | 16.91 | 45.28 | 41.74 | 60.02 | 56.20 | 48.28 | 44.56 |
| Gemma-7B | FedIT | 71.54 | 47.59 | 28.89 | 19.81 | 53.42 | 44.51 | 65.73 | 55.32 | 54.90 | 41.81 |
| | FFA-LoRA | 72.32 | 46.12 | 28.70 | 18.19 | 53.61 | 43.12 | 65.40 | 54.95 | 55.01 | 40.59 |
| | FedSA-LoRA | 75.93 | 35.20 | 30.34 | 14.92 | 55.25 | 38.61 | 68.22 | 48.42 | 57.44 | 34.29 |
| | FLoRA | 71.35 | 65.04 | 29.61 | 22.76 | 53.43 | 44.80 | 64.42 | 59.42 | 54.70 | 48.01 |
| | FedEx-LoRA | 71.48 | 67.28 | 28.75 | 23.91 | 54.12 | 46.89 | 64.84 | 60.63 | 54.80 | 49.68 |
| | FLoRA-NA | 72.44 | 69.42 | 29.37 | 25.91 | 54.80 | 49.72 | 65.40 | 62.10 | 55.50 | 51.79 |

methods improve test accuracy as data heterogeneity decreases (i.e., larger $\alpha$). Notably, FLoRA-NA consistently achieves superior and stable performance across different levels of heterogeneity, indicating its robustness under non-IID conditions.

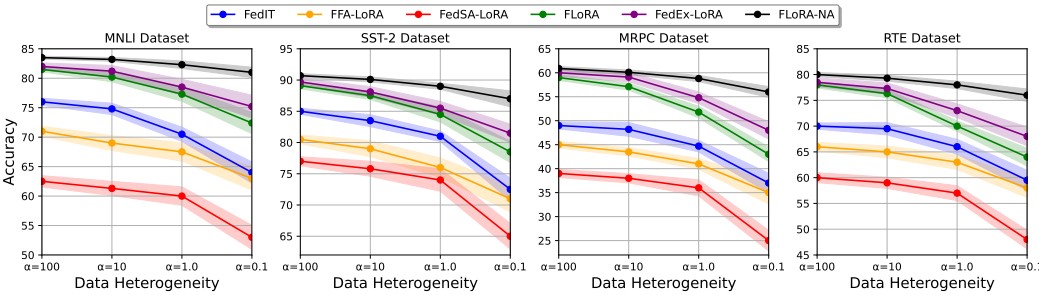

Figure 3: Performance w.r.t data heterogeneity $\alpha$ for four datasets.

**Varying client number.** We show the specific performance of the methods under various size of FL system in Table 4. Notably, FLoRA-NA consistently outperforms all baselines across MNLI, QNLI, and SST-2, regardless of the number of clients. It can also be seen that FedSA-LoRA performance drops significantly as the number of client increases, highlighting the overfitting effect caused by personalization.

Table 4: Global model performance comparison on the QNLI, SST2, and MNLI-m tasks with different number of clients. We apply full participation for the FL system with 10, 20 and 100 clients.

| Method | MNLI | | | QNLI | | | SST-2 | | |
|---|---|---|---|---|---|---|---|---|---|
| | 10 clients | 20 clients | 100 clients | 10 clients | 20 clients | 100 clients | 10 clients | 20 clients | 100 clients |
| FedIT | 71.54 | 52.62 | 21.39 | 72.84 | 56.07 | 24.48 | 79.56 | 58.33 | 26.57 |
| FedSA-LoRA | 58.33 | 39.22 | 13.69 | 63.47 | 43.02 | 15.80 | 72.21 | 45.76 | 18.78 |
| FLoRA | 74.11 | 64.01 | 36.76 | 78.51 | 67.86 | 35.14 | 83.05 | 75.42 | 38.04 |
| FedEx-LoRA | 78.05 | 70.13 | 43.61 | 80.31 | 71.94 | 46.67 | 84.92 | 74.62 | 50.96 |
| FLoRA-NA | 81.13 | 74.40 | 48.62 | 84.59 | 76.81 | 53.34 | 87.05 | 78.05 | 56.92 |

## 5.5 COMMUNICATION AND COMPUTATION EFFICIENCY.

We compare FLoRA-NA with baselines in terms of system efficiency. As shown in Table 5, FLoRA-NA maintains a comparable number of trainable parameters, communication overhead, and per-round execution time with light-weight FedIT, while achieving a significantly faster convergence owing to its accurate aggregation strategy. We also highlight two important observations that highlight the superiority of FLoRA-NA: (1) FLoRA converges significantly slower than the naive FedIT. This is due to the re-initialization of $A$ and $B$ matrices at each communication round that introduces additional noise[1]. Also, $B$ is initialized with zeros, leading to a necessary warm-up phase that hinders efficiency when re-initialization is repeated continuously. (2) FedEx-LoRA suffers from an extremely high communication cost due to residual error broadcast from server to all clients, which is difficult to apply on real-world FL system with limited communication bandwidth.

Table 5: System efficiency of each method on MNLI and STS-B datasets. Communication round denotes the number of communication rounds to reach the predefined target performance (95% of the results reported in Table 2).

| Method | Trainable Parm. per Client | Per-round Communicated Parm. | | Per-round Execution Time | | Communication Round | |
|---|---|---|---|---|---|---|---|
| | | C→S | S→C | MNLI | STS-B | MNLI | STS-B |
| FedIT | 1.83M | 0.78M | 0.78M | 42s | 25s | 412 | 250 |
| FedDPA-LoRA | 2.62M | 0.78M | 0.78M | 75s | 52s | 409 | 252 |
| FFA-LoRA | 1.44M | 0.39M | 0.39M | 38s | 22s | 387 | 224 |
| FedSA-LoRA | 1.83M | 0.39M | 0.39M | 41s | 24s | 366 | 209 |
| FLoRA | 1.83M | 0.78M | 7.8M | 58s | 43s | 492 | 313 |
| FedEx-LoRA | 1.83M | 0.78M | 101.64M | 55s | 39s | 314 | 184 |
| FLoRA-NA | 1.83M | 0.78M | 0.78M | 43s | 26s | 283 | 161 |

## 5.6 COMPATIBILITY WITH COMPRESSION METHODS

As discussed in Section 3, compression techniques such as sparsification and quantization can be seamlessly integrated into the aggregated LoRA matrices of FLoRA-NA to further reduce communication overhead. To evaluate this compatibility, we conduct experiments across different quantization and sparsification levels. The results, presented in Table 6, demonstrate that FLoRA-NA can incorporate compression methods, achieving substantial communication savings with only a slight trade-off in accuracy.

Table 6: FLoRA-NA performance under various bit compression methods on the MNLI dataset using the RoBERTa-large model. The training settings are the same as in Section 5.3.

| Compression method | Per-round Communicated Parm. | | Accuracy |
|---|---|---|---|
| | C→S | S→C | |
| None (32 bits) | 0.78M | 0.78M | 81.13% |
| Half-precision (16 bits) | 0.39M | 0.39M | 79.91% |
| Quantization (16 bits) | 0.39M | 0.39M | 77.54% |
| Quantization (8 bits) | 0.195M | 0.195M | 73.24% |
| Sparsification (50%) | 0.39M | 0.39M | 79.85% |
| Sparsification (75%) | 0.195M | 0.195M | 74.42% |

## 6 CONCLUSION

In this work, we revisit existing federated fine-tuning methods and identify key challenges related to both global-local generalization and communication efficiency. To address these practical challenges and expand the applicability of federated fine-tuning, we propose FLoRA-NA, a novel approach designed to bridge the gap between global generalization and local personalization while maintaining communication efficiency. This is achieved by enabling accurate on-server aggregation of LoRA matrices. Extensive experimental results demonstrate that FLoRA-NA consistently outperforms state-of-the-art methods across both homogeneous and heterogeneous LoRA settings. Furthermore, FLoRA-NA preserves the communication efficiency of the standard federated LoRA approach, in which LoRA matrices are directly averaged. These promising results provide valuable insights and establish a foundation for future research on lightweight and accurate federated fine-tuning of foundation models.

---

[1]This can be found in line 280 of the official code of FLoRA at: https://github.com/ziyaow1010/FederatedLLM/blob/main/main.py.

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

# A  PROOF ON THEORETICAL CONVERGENCE

In our theoretical analysis, we extend and refine the convergence frameworks developed in prior studies (T Dinh et al., 2020; Jhunjhunwala et al., 2023). While those works provide a foundation for analyzing the behavior of federated learning algorithms, our contribution lies in uncovering a deeper connection between conventional FL and its counterparts that employ LoRA. Specifically, we establish this connection through the notion of divergence between two forms of aggregated parameters: the conventional aggregation of full-resolution model weights, denoted by $\bar{W}$, and the aggregation obtained under the LoRA decomposition, expressed as $\hat{W} = \hat{B}\hat{A}$. By formalizing this divergence, we are able to highlight how LoRA-based federated algorithms deviate from the baseline FL approach in terms of parameter aggregation. The discussion leading to the definition of this divergence and its theoretical implications is provided in Section 2.2, which serves as the foundation for the subsequent analysis.

## A.1  KEY ASSUMPTIONS AND LEMMAS

**Assumption 2** ($L$-smoothness). *For each client $u \in [U]$, the local objective $\mathcal{L}_u(W)$ is differentiable and $L$-smooth, i.e.,*

$$\|\nabla\mathcal{L}_u(W) - \nabla\mathcal{L}_u(W')\| \le L\|W - W'\|, \quad \forall W, W' \in \mathbb{R}^d.$$

**Assumption 3** (Bounded data heterogeneity at optimum). *At the global optimum $W^*$, the average squared norm of client gradients is bounded:*

$$\frac{1}{U}\sum_{u=1}^{U}\|\nabla\mathcal{L}_u(W^*)\|^2 \le \sigma_*^2.$$

**Lemma 1** (Jensen's Inequality). *For any $\mathbf{a}_u \in \mathbb{R}^d$, for $u \in \{1, 2, \ldots, U\}$:*

$$\left\|\frac{1}{U}\sum_{u=1}^{U}\mathbf{a}_u\right\|^2 \le \frac{1}{U}\sum_{u=1}^{U}\left\|\mathbf{a}_u\right\|^2, \tag{7}$$

$$\left\|\sum_{u=1}^{U}\mathbf{a}_u\right\|^2 \le U\sum_{u=1}^{U}\left\|\mathbf{a}_u\right\|^2. \tag{8}$$

**Lemma 2** (Gradient bound via Bregman divergence). *If $F$ is smooth and convex, then*

$$\|\nabla\mathcal{L}(W) - \nabla\mathcal{L}(W')\|^2 \le 2L(\mathcal{L}(W) - \mathcal{L}(W') - \langle\nabla\mathcal{L}(W'), W - W'\rangle). \tag{9}$$

**Lemma 3** (Co-coercivity of convex smooth function). *If $\mathcal{L}$ is $L$-smooth and convex then,*

$$\langle\nabla\mathcal{L}(W) - \nabla\mathcal{L}(W'), W - W'\rangle \ge \frac{1}{L}\|\nabla\mathcal{L}(W) - \nabla\mathcal{L}(W')\|^2. \tag{10}$$

As a direct consequence, for the global minimizer $W^*$,

$$(\nabla\mathcal{L}(W), W - W^*) \ge \frac{1}{L}\|\nabla\mathcal{L}(W)\|^2. \tag{11}$$

## A.2  PROOF OF LEMMA 4

**Lemma 4.** *Consider local objectives $\mathcal{L}_u(W)$ that are convex and $L$-smooth for all $u \in [U]$, and assume that $W^*$ is a common minimizer of these functions. If each client performs full-batch gradient descent with stepsize $\eta \le 1/L$, then for any communication round $t$ and any number of local steps $E \ge 1$, the following holds:*

$$\frac{1}{U}\sum_{u=1}^{U}\left\|W_u^{(t,E)} - W^*\right\|^2 \le \left\|W^{(t)} - W^*\right\|^2. \tag{12}$$

Let $\mathcal{L}_u(W)$ denote the local objective for client $u$ and $W^*$ the global minimizer. Under the over-parameterization assumption, $W^*$ is also a minimizer of each $\mathcal{L}_u(W)$. Consider the local update at client $u$:

$$W_u^{(t,e+1)} = W_u^{(t,e)} - \eta \nabla \mathcal{L}_u\big(W_u^{(t,e)}\big). \tag{13}$$

We examine the distance to the global minimizer:

$$\left\| W_u^{(i,E)} - W^* \right\|^2 = \left\| W_u^{(t,k-1)} - \eta \nabla \mathcal{L}(W_u^{(t,k-1)}) - W^* \right\|^2 \tag{14}$$

$$= \left\| W_u^{(t,k-1)} - W^* \right\|^2 - 2\eta \Big\langle \nabla \mathcal{L}(W_u^{(t,k-1)}), W_u^{(t,k-1)} - W^* \Big\rangle$$

$$+ \eta^2 \left\| \nabla \mathcal{L}(W_u^{(t,k-1)}) \right\|^2 \tag{15}$$

$$\leq \left\| W_u^{(t,k-1)} - W^* \right\|^2 - \frac{2\eta}{L} \left\| \nabla \mathcal{L}(W_u^{(t,k-1)}) \right\|^2 + \eta^2 \left\| \nabla \mathcal{L}(W_u^{(t,k-1)}) \right\|^2 \tag{16}$$

$$\leq \left\| W_u^{(t,k-1)} - W^* \right\|^2 - \frac{\eta}{L} \left\| \nabla \mathcal{L}(W_u^{(t,k-1)}) \right\|^2, \tag{17}$$

where (20) follows from (17) and (21) follows from $\eta \leq 1/L$.

Summing over $e = 0$ to $E - 1$, we have

$$\left\| W_u^{(t,E)} - W^* \right\|^2 \leq \left\| W^{(i)} - W^* \right\|^2 - \frac{\eta}{L} \sum_{e=0}^{E-1} \left\| \nabla \mathcal{L}(W_u^{(i,E)}) \right\|^2. \tag{18}$$

Finally, averaging over all clients yields

$$\frac{1}{U} \sum_{u=1}^{U} \left\| W_u^{(t,E)} - W^* \right\|^2 \leq \left\| W^{(i)} - W^* \right\|^2 - \frac{\eta}{ML} \sum_{u=1}^{U} \sum_{e=0}^{E-1} \left\| \nabla \mathcal{L}(W_u^{(i,E)}) \right\|^2 \tag{19}$$

$$\leq \left\| W^{(i)} - W^* \right\|^2. \tag{20}$$

which completes the proof. $\qquad\square$

### A.3 PROOF ON LEMMA 5

**Lemma 5** (Bounding client aggregate gradients).

$$\frac{1}{U} \sum_{u=1}^{U} \sum_{e=0}^{E-1} \left\| \nabla \mathcal{L}_u(W_u^{(i,E)}) \right\|^2 \leq \frac{3L^2}{U} \sum_{u=1}^{U} \sum_{e=0}^{E-1} \left\| W_u^{(i,E)} - W^{(i)} \right\|^2 + 6EL\big(\mathcal{L}(W^{(i)}) - \mathcal{L}(W^*)\big) + 3E\sigma_*^2. \tag{21}$$

**Proof.**

$$\frac{1}{U} \sum_{u=1}^{U} \sum_{e=0}^{E-1} \left\| \nabla \mathcal{L}_u(W_u^{(i,E)}) \right\|^2$$

$$= \frac{1}{U} \sum_{u=1}^{U} \sum_{e=0}^{E-1} \left\| \nabla \mathcal{L}_u(W_u^{(i,E)}) - \nabla \mathcal{L}_u(W^{(i)}) + \nabla \mathcal{L}_u(W^{(i)}) - \nabla \mathcal{L}_u(W^*) + \nabla \mathcal{L}_u(W^*) \right\|^2 \tag{22}$$

$$\leq \frac{3}{U} \sum_{u=1}^{U} \sum_{e=0}^{E-1} \left\| \nabla \mathcal{L}_u(W_u^{(i,E)}) - \nabla \mathcal{L}_u(W^{(i)}) \right\|^2 + \frac{3}{U} \sum_{u=1}^{U} \sum_{e=0}^{E-1} \left\| \nabla \mathcal{L}_u(W^{(i)}) - \nabla \mathcal{L}_u(W^*) \right\|^2$$

$$+ \frac{3}{U} \sum_{u=1}^{U} \sum_{e=0}^{E-1} \left\| \nabla \mathcal{L}_u(W^*) \right\|^2 \tag{23}$$

$$\leq \frac{3L^2}{U} \sum_{u=1}^{U} \sum_{e=0}^{E-1} \left\| W_u^{(i,E)} - W^{(i)} \right\|^2 + 6EL\big(\mathcal{L}(W^{(i)}) - \mathcal{L}(W^*)\big) + 3E\sigma_*^2. \tag{24}$$

The first term in (28) follows from $L$-smoothness of $\mathcal{L}_u(W)$, the second term follows from Lemma 3, and the third term follows from bounded noise at optimum. □

## A.4 PROOF OF LEMMA 6

**Lemma 6** (Bounding client drift).

$$\frac{1}{U}\sum_{u=1}^{U}\sum_{e=0}^{E-1}\left\|W^{(i)}-W_u^{(i,E)}\right\|^2 \leq 12\eta^2 E^2(E-1)L\big(\mathcal{L}(W^{(i)})-\mathcal{L}(W^*)\big)+6\eta^2 E^2(E-1)\sigma_*^2. \tag{25}$$

**Proof.**

$$\frac{1}{U}\sum_{u=1}^{U}\sum_{e=0}^{E-1}\left\|W^{(i)}-W_u^{(i,E)}\right\|^2 = \eta^2\frac{1}{U}\sum_{u=1}^{U}\sum_{e=0}^{E-1}\left\|\sum_{l=0}^{k-1}\nabla\mathcal{L}_u(W_u^{(t,l)})\right\|^2 \tag{26}$$

$$\leq \eta^2\frac{1}{U}\sum_{u=1}^{U}\sum_{e=0}^{E-1}e\sum_{l=0}^{k-1}\left\|\nabla\mathcal{L}_u(W_u^{(t,l)})\right\|^2 \tag{27}$$

$$\leq \eta^2 E(E-1)\frac{1}{U}\sum_{u=1}^{U}\sum_{e=0}^{E-1}\left\|\nabla\mathcal{L}_u(W_u^{(i,E)})\right\|^2 \tag{28}$$

$$\leq 3\eta^2 E(E-1)L^2\frac{1}{U}\sum_{u=1}^{U}\sum_{e=0}^{E-1}\left\|W_u^{(i,E)}-W^{(i)}\right\|^2$$
$$+ 6\eta^2 E^2(E-1)L\big(\mathcal{L}(W^{(i)})-\mathcal{L}(W^*)\big)+3\eta^2 E^2(E-1)\sigma_*^2 \tag{29}$$

$$\leq \frac{1}{2U}\sum_{u=1}^{U}\sum_{e=0}^{E-1}\left\|W^{(i)}-W_u^{(i,E)}\right\|^2$$
$$+ 6\eta^2 E^2(E-1)L\big(\mathcal{L}(W^{(i)})-\mathcal{L}(W^*)\big)+3\eta^2 E^2(E-1)\sigma_*^2, \tag{30}$$

where (33) uses Lemma 5 and (34) uses $\eta \leq \frac{1}{6EL}$.

Therefore, we have

$$\frac{1}{U}\sum_{u=1}^{U}\sum_{e=0}^{E-1}\left\|W^{(i)}-W_u^{(i,E)}\right\|^2 \leq 12\eta^2 E^2(E-1)L\big(\mathcal{L}(W^{(i)})-\mathcal{L}(W^*)\big)+6\eta^2 E^2(E-1)\sigma_*^2. \tag{31}$$

□

## A.5 PROOF ON CONVERGENCE OF CONVEX OBJECTIVES

We introduce the following auxiliary variables, which will be utilized throughout the proof.

$$\text{Aggregate Client Gradient:} \quad h_u^{(i)} = \sum_{e=0}^{E-1} \nabla \mathcal{L}_u(W_u^{(i,E)}). \tag{36}$$

We also define $\bar{h}^{(i)} = \frac{1}{U} \sum_{u=1}^{U} h_u^{(i)}$, and $\nabla \hat{W}$ is the joint gradient computed by aggregating the local LoRA matrices via federated LoRA algorithms.

**Definition 2.** *Denote* $\varrho = \mathbb{E}\|\nabla \hat{W} - \nabla \bar{W}\| = \mathbb{E}\|\nabla \hat{W} - h^{(i)}\|$, *we have the followings:*

$$\nabla \hat{W}^{(i)} = \bar{h}^{(i)} + \varphi, \quad s.t. \quad \varphi \sim \mathcal{N}(0, \varrho^2), \quad and \quad \varphi \in \mathbb{R}^d \tag{32}$$

Recall that the update of the global model can be written as $W^{(i+1)} = W^{(i)} - \eta \bar{h}^{(i)}$.

We have

$$\left\| W^{(i+1)} - W^* \right\|^2$$

$$= \left\| W^{(i)} - \eta \nabla \hat{W}^{(i)} - W^* \right\|^2 \tag{33}$$

$$= \left\| W^{(i)} - W^* \right\|^2 - 2\eta \left\langle W^{(i)} - W^*, \nabla \hat{W}^{(i)} \right\rangle + \eta^2 \left\| \nabla \hat{W}^{(i)} \right\|^2 \tag{34}$$

$$\overset{(a)}{=} \left\| W^{(i)} - W^* \right\|^2 - 2\eta \left\langle W^{(i)} - W^*, \nabla \hat{W}^{(i)} \right\rangle + \eta^2 \left\| \bar{h}^{(i)} + \varphi \right\|^2 \tag{35}$$

$$\leq \left\| W^{(i)} - W^* \right\|^2 \underbrace{- 2\eta \left\langle W^{(i)} - W^*, \nabla \hat{W}^{(i)} \right\rangle}_{Q_1} + \underbrace{\eta^2 \frac{1}{U} \sum_{u=1}^{U} \left\| h_u^{(i)} \right\|^2}_{Q_2} + \eta^2 \frac{1}{U} \sum_{u=1}^{U} \|\varphi\|^2, \tag{36}$$

where (a) holds due to (2). Bounding $Q_1, Q_2$, we have

$$Q_2 = \frac{1}{U} \sum_{u=1}^{U} \|h_u^{(i)}\|^2 \tag{37}$$

$$= \frac{1}{U} \sum_{u=1}^{U} \left\| \sum_{e=0}^{E-1} \nabla \mathcal{L}_u(W_u^{(i,E)}) \right\|^2 \tag{38}$$

$$\leq \frac{E}{U} \sum_{u=1}^{U} \sum_{e=0}^{E-1} \left\| \nabla \mathcal{L}_u(W_u^{(i,E)}) \right\|^2 \tag{39}$$

$$\leq \frac{3EL^2}{U} \sum_{u=1}^{U} \sum_{e=0}^{E-1} \left\| W_u^{(i,E)} - W^{(i)} \right\|^2 + 6E^2 L\big(\mathcal{L}(W^{(i)}) - \mathcal{L}(W^*)\big) + 3E^2 \sigma_*^2, \tag{40}$$

where (43) uses Jensen's inequality and (44) follows from Lemma 5. Similarly,

$$Q_1 = \frac{1}{U} \sum_{u=1}^{U} \left\langle W^{(i)} - W^*, \nabla \hat{W}^{(i)} \right\rangle = \frac{1}{U} \sum_{u=1}^{U} \left\langle W^{(i)} - W^*, \bar{h}^{(i)} + \varphi \right\rangle \tag{41}$$

$$= \frac{1}{U} \sum_{u=1}^{U} \left\langle W^{(i)} - W^*, h_u^{(i)} \right\rangle + \frac{1}{U} \sum_{u=1}^{U} \left\langle W^{(i)} - W^*, h_u^{(i)} - \frac{1}{U} \sum_{u=1}^{U} B_u \sum_{u=1}^{U} A_u \right\rangle$$

$$= \frac{1}{U} \sum_{u=1}^{U} \sum_{e=0}^{E-1} \left\langle W^{(i)} - W^*, \nabla \mathcal{L}_u(W_u^{(i,E)}) \right\rangle. \tag{42}$$

We have:

$$\left\langle W^{(i)} - W^*, \nabla \mathcal{L}_u(W_u^{(i,E)}) \right\rangle = \left\langle W^{(i)} - W^{(i,E)}, \nabla \mathcal{L}_u(W_u^{(i,E)}) \right\rangle + \left\langle W^{(i,E)} - W^*, \nabla \mathcal{L}_u(W_u^{(i,E)}) \right\rangle \tag{43}$$

From $L$-smoothness of $\mathcal{L}_u$, we have:

$$\left\langle W^{(i)} - W^{(i,E)}, \nabla\mathcal{L}_u(W_u^{(i,E)})\right\rangle \leq \mathcal{L}_u(W^{(i)}) - \mathcal{L}_u(W^{(i,E)}) - \frac{L}{2}\|W^{(i)} - W_u^{(i,E)}\|^2 \quad (44)$$

From convexity of $\mathcal{L}_u$, we have:

$$\left\langle W^{(i,E)} - W^*, \nabla\mathcal{L}_u(W_u^{(i,E)})\right\rangle \leq \mathcal{L}_u(W^{(i,E)}) - \mathcal{L}_u(W^*) \quad (45)$$

We expand and apply $L$-smoothness and convexity to derive:

$$\left\langle W^{(i)} - W^*, \nabla\mathcal{L}_u(W_u^{(i,E)})\right\rangle \geq \mathcal{L}_u(W^{(i)}) - \mathcal{L}_u(W^*) - \frac{L}{2}\|W^{(i)} - W_u^{(i,E)}\|^2. \quad (46)$$

Substituting (46) into (41):

$$Q_1 \geq E\Big(\mathcal{L}(W^{(i)}) - \mathcal{L}(W^*)\Big) - \frac{L}{2U}\sum_{u=1}^{U}\sum_{e=0}^{E-1}\|W^{(i)} - W_u^{(i,E)}\|^2. \quad (47)$$

Note that our main contribution lays in (35) and (36), where we decompose the gradients into gradient with noise $\varrho$. The noise is induced by the divergence between the ideal low-rank matrix decomposition and the federated LoRA algorithm as introduced in Definition 1.

Substituting bounds for $T_1$ and $T_2$ into (36), we get:

$$\|W^{(i+1)} - W^*\|^2 \leq \|W^{(i)} - W^*\|^2 - 2\eta E(1 - 3\eta EL)(\mathcal{L}(W^{(i)}) - \mathcal{L}(W^*)) + 3\eta^2 E^2\sigma_*^2$$

$$+ \big(3\eta^2 EL^2 + \eta L\big)\frac{1}{U}\sum_{u=1}^{U}\sum_{e=0}^{E-1}\|W^{(i)} - W_u^{(i,E)}\|^2 + \eta^2\frac{1}{U}\sum_{u=1}^{U}\|\varphi\|^2$$

$$\leq \|W^{(i)} - W^*\|^2 - \eta E(\mathcal{L}(W^{(i)}) - \mathcal{L}(W^*)) + 3\eta^2 E^2\sigma_*^2$$

$$+ 2\eta L\frac{1}{U}\sum_{u=1}^{U}\sum_{e=0}^{E-1}\|W^{(i)} - W_u^{(i,E)}\|^2 + \eta^2\frac{1}{U}\sum_{u=1}^{U}\|\varphi\|^2 \quad (48)$$

$$\overset{(1)}{\leq} \|W^{(i)} - W^*\|^2 - \eta E(\mathcal{L}(W^{(i)}) - \mathcal{L}(W^*)) + 3\eta^2 E^2\sigma_*^2$$

$$+ 24\eta_l^3 E^2(E - 1)L^2(\mathcal{L}(W^{(i)}) - \mathcal{L}(W^*)) + 12\eta_l^3 E^2(E - 1)L\sigma_*^2$$

$$+ \eta^2\frac{1}{U}\sum_{u=1}^{U}\|\varphi\|^2 \quad (49)$$

$$\overset{(2)}{\leq} \|W^{(i)} - W^*\|^2 - \frac{\eta E}{3}(\mathcal{L}(W^{(i)}) - \mathcal{L}(W^*)) + 3\eta^2 E^2\sigma_*^2$$

$$+ 12\eta_l^3 E^2(E - 1)L\sigma_*^2 + \eta^2\frac{1}{U}\sum_{u=1}^{U}\|\varphi\|^2. \quad (50)$$

where both (49) and (50) use $\eta \leq \frac{1}{6EL}$, and (48) uses Lemma 6.

Averaging over all rounds and rearranging terms, we finally obtain:

$$\mathcal{L}(W^{(i)}) - \mathcal{L}(W^*) \leq \frac{3\|W^{(0)} - W^*\|^2}{\sum_{i=0}^{R-1}\eta E} + 9\eta E\sigma_*^2 + 36\eta^2 E(E - 1)L\sigma_*^2 + 3\eta^2 kd\varrho. \quad (51)$$

This implies

$$\mathcal{L}(W^{(i)}) - \mathcal{L}(W^*) \leq \mathcal{O}\left(\frac{\|W^{(0)} - W^*\|^2}{\sum_{i=0}^{R-1}\eta E}\right) + \mathcal{O}\left(\eta E\sigma_*^2\right) + \mathcal{O}\left(\eta^2 E(E - 1)L\sigma_*^2\right) + \mathcal{O}\left(\eta^2 kd\varrho\right). \quad (52)$$

This completes the proof of Theorem 1. $\qquad\square$

### A.6 Proof on Convergence of Non-Convex Objectives

The update of the global model can be written as follows,

$$W^{(i+1)} = W^{(i)} - \eta E \nabla \hat{W}^{(i)}. \tag{53}$$

Now using the Lipschitz-smoothness assumption we have,

$$\mathcal{L}(W^{(i+1)}) - \mathcal{L}(W^{(i)}) \leq -\eta E \left\langle \nabla \mathcal{L}(W^{(i)}), \nabla \hat{W}^{(i)} \right\rangle + \frac{\eta^2 E^2 L}{2} \left\| \nabla \hat{W}^{(i)} \right\|^2 \tag{54}$$

$$\leq -\eta E \left\langle \nabla \mathcal{L}(W^{(i)}), \bar{h}^{(i)} + \varphi \right\rangle + \frac{\eta^2 E^2 L}{2} \left\| \nabla \bar{h}^{(i)} + \varphi \right\|^2 \tag{55}$$

$$\overset{(a)}{\leq} -\eta E \left\langle \nabla \mathcal{L}(W^{(i)}), \bar{h}^{(i)} \right\rangle + \frac{\eta^2 E^2 L}{2U} \sum_{u=1}^{U} \left\| h_u^{(i)} \right\|^2 + \frac{\eta^2 E^2 L}{2U} \sum_{u=1}^{U} \| \varphi \|^2. \tag{56}$$

The inequality $(a)$ holds due to $\varphi = \bar{h}^{(i)} - \hat{h}^{(i)}$, and the difference is relatively small compared to $\nabla \mathcal{L}(W^{(i)})$, and thus, $\varphi$ is nearly orthogonal with $\nabla \mathcal{L}(W^{(i)})$, and $\left\langle \nabla \mathcal{L}(W^{(i)}), \bar{h}^{(i)} + \varphi \right\rangle \ll 1$. Following the proof of (Jhunjhunwala et al., 2023) we have,

$$\mathcal{L}(W^{(i+1)}) - \mathcal{L}(W^{(i)}) \leq -\eta E \left\langle \nabla \mathcal{L}(W^{(i)}), \bar{h}^{(i)} \right\rangle + \frac{\eta^2 E^2 L}{2U} \sum_{u=1}^{U} \left\| h_u^{(i)} \right\|^2 + \frac{\eta^2 E^2 L}{2U} \sum_{u=1}^{U} \| \varphi \|^2$$

$$\leq -\eta E \underbrace{\left\langle \nabla \mathcal{L}(W^{(i)}), \bar{h}^{(i)} \right\rangle}_{T_1} + \frac{\eta^2 E^2 L}{2U} \underbrace{\sum_{u=1}^{U} \left\| h_u^{(i)} \right\|^2}_{T_2} + \frac{\eta^2 E^2 L}{2} k d\varrho. \tag{57}$$

**Bounding $T_1$**

We have,

$$T_1 = \left\langle \nabla \mathcal{L}(W^{(i)}), \frac{1}{U} \sum_{i=1}^{U} h_u^{(i)} \right\rangle \tag{58}$$

$$= \frac{1}{2} \left\| \nabla \mathcal{L}(W^{(i)}) \right\|^2 + \frac{1}{2} \left\| \frac{1}{U} \sum_{u=1}^{U} h_u^{(i)} \right\|^2 - \frac{1}{2} \left\| \nabla \mathcal{L}(W^{(i)}) - \frac{1}{U} \sum_{u=1}^{U} h_u^{(i)} \right\|^2 \tag{59}$$

$$\geq \frac{1}{2} \left\| \nabla \mathcal{L}(W^{(i)}) \right\|^2 - \frac{1}{2U} \sum_{u=1}^{U} \left\| \nabla \mathcal{L}_u(W^{(i)}) - h_u^{(i)} \right\|^2, \tag{60}$$

where (59) uses $\langle a, b \rangle = \frac{1}{2} \|a\|^2 + \frac{1}{2} \|b\|^2 - \frac{1}{2} \|a - b\|^2$ and (60) uses Jensen's inequality and the definition of the global objective function $\mathcal{L}$.

**Bounding $T_2$**

We have,

$$T_2 = \frac{1}{U} \sum_{u=1}^{U} \left\| h_u^{(i)} \right\|^2 \tag{61}$$

$$= \frac{1}{U} \sum_{u=1}^{U} \left\| h_u^{(i)} - \nabla \mathcal{L}_u(W^{(i)}) + \nabla \mathcal{L}_u(W^{(i)}) - \nabla \mathcal{L}(W^{(i)}) + \nabla \mathcal{L}(W^{(i)}) \right\|^2 \tag{62}$$

$$\leq \frac{3}{U} \sum_{u=1}^{U} \left( \left\| h_u^{(i)} - \nabla \mathcal{L}_u(W^{(i)}) \right\|^2 + \left\| \nabla \mathcal{L}_u(W^{(i)}) - \nabla \mathcal{L}(W^{(i)}) \right\|^2 + \left\| \nabla \mathcal{L}(W^{(i)}) \right\|^2 \right) \tag{63}$$

$$\leq \frac{3}{U} \sum_{u=1}^{U} \left\| h_u^{(i)} - \nabla \mathcal{L}_u(W^{(i)}) \right\|^2 + 3\sigma_g^2 + 3 \left\| \nabla \mathcal{L}(W^{(i)}) \right\|^2, \tag{64}$$

where (63) uses Jensen's inequality, (64) uses bounded data heterogeneity assumption. The bound $T_1$ follows (Jhunjhunwala et al., 2023). The bound for $T_1$ follows a similar technique as in (Wang et al., 2020). Substituting the $T_1$ and $T_2$ bounds into (57), we have,

$$\mathcal{L}(W^{(i+1)}) - \mathcal{L}(W^{(i)})$$

$$\leq -\eta E\left( \frac{1}{2} \left\| \nabla\mathcal{L}(W^{(i)}) \right\|^2 + \frac{1}{2U} \sum_{u=1}^{U} \left\| \nabla\mathcal{L}_u(W^{(i)}) - h_u^{(i)} \right\|^2 \right. \tag{65}$$

$$\left. + \frac{\eta E L}{2}\left( 3\sigma_g^2 + 3\left\| \nabla\mathcal{L}(W^{(i)}) \right\|^2 + \frac{3}{U}\sum_{u=1}^{U}\left\| h_u^{(i)} - \nabla\mathcal{L}_u(W^{(i)}) \right\|^2 \right) \right) + \frac{\eta^2 E^2 L}{2}kd\varrho$$

$$\leq -\eta E\left( \frac{1}{4} \left\| \nabla\mathcal{L}(W^{(i)}) \right\|^2 + \frac{1}{U} \sum_{u=1}^{U} \left\| \nabla\mathcal{L}_u(W^{(i)}) - h_u^{(i)} \right\|^2 + 3\eta E L\sigma_g^2 \right) + \frac{\eta^2 E^2 L}{2}kd\varrho \tag{66}$$

$$\leq -\eta E\left( \frac{1}{8} \left\| \nabla\mathcal{L}(W^{(i)}) \right\|^2 + 3\eta E L\sigma_g^2 + 5\eta^2 L^2 E(E-1)\sigma_g^2 + \frac{\eta E L}{2}kd\varrho \right), \tag{67}$$

where (66) uses $\eta \leq \frac{1}{6EL}$ and (67) uses (Jhunjhunwala et al., 2023, Lemma 7). Thus rearranging terms and averaging over all rounds we have,

$$\frac{\sum_{i=0}^{R-1}\left\| \nabla\mathcal{L}(W^{(i)}) \right\|^2}{\sum_{i=0}^{R-1}} \leq \frac{8(\mathcal{L}(W^{(0)}) - F^*)}{\sum_{i=0}^{R-1}\eta E} + 40\eta^2 L^2 E(E-1)\sigma_g^2 + 24\eta L E\sigma_g^2 + 4\eta E L kd\varrho. \tag{68}$$

This implies

$$\min_{i\in[R]}\left\| \hat{B}^{(i)}\hat{A}^{(i)} \right\|^2 \leq \mathcal{O}\left( \frac{(\mathcal{L}(W^{(0)}) - \mathcal{L}^*)}{\sum_{i=0}^{R-1}\eta E} \right) + \mathcal{O}\left( \eta^2 L^2 E(E-1)\sigma_g^2 \right) + \mathcal{O}\left( \eta L E\sigma_g^2 \right) + \mathcal{O}\left( kd\varrho \right). \tag{69}$$

This completes the proof of Theorem 4. $\qquad\square$

## B  RELATED WORKS

### B.1  FEDERATED FINE-TUNING WITH PRE-TRAINED LANGUAGE MODELS

Pre-trained LLMs have largely exhausted publicly available data and now require access to private, domain-specific datasets to further advance. However, this data is typically distributed across multiple parties, each possessing only a small amount that is insufficient for independently fine-tuning large models. Moreover, these parties are often restricted from sharing their data directly with others due to privacy or regulatory constraints. A popular solution to this problem is federated learning (Huang et al., 2024; Tian et al., 2022; Kairouz et al., 2021; McMahan et al., 2017), which enables several clients to collaboratively fine-tune LLMs by sharing their local model updates without exposing their raw data. Despite this advantage, the ever-growing size of LLMs creates significant communication costs in federated settings (Zhang et al., 2024b; Hu et al., 2022; Deng et al., 2024). Moreover, clients typically have limited computational power and memory, which makes local fine-tuning of massive LLMs difficult or even infeasible. To address these issues, many Parameter-Efficient Fine-Tuning (PEFT) techniques have been proposed. These methods introduce a small number of extra trainable parameters to adapt the model, while keeping most of the original pre-trained weights frozen. For example, some works add small trainable modules called adapters into each layer of the network (Zhang et al., 2024b; Pfeiffer et al., 2021; Houlsby et al., 2019; Rebuffi et al., 2017). Others modify the architecture by attaching additional trainable vectors to the inputs or hidden layers (Deng et al., 2024; Li et al., 2024; Qiu et al., 2024; Li & Liang, 2021). Another prominent line of work (Huang et al., 2025; Liu et al., 2024; Sun et al., 2024; Kalajdzievski, 2023; Hu et al., 2022) introduces LoRA, which uses a low-rank decomposition of weight updates into two smaller matrices, $A$ and $B$, to efficiently approximate full-rank adaptations during fine-tuning. Among these, LoRA is now widely adopted, reaching performance close to full fine-tuning while only need updating less than $1\%$ of the model's parameters.

### B.2  LORA IN FEDERATED LEARNING

Low-Rank Adaptation (LoRA) is increasingly adopted into FL due to its light-weight update but competitive performance compare to traditional full model fine-tuning. FedIT (Zhang et al., 2024a) and pFedLoRA (Yi et al., 2023) are the pioneers introduced applying LoRA in federated scenario in the most naive way. (Liu et al., 2025) introduced DP-LoRA, which guarantees differential privacy in federated learning for large language models while keeping communication costs low. (Yang et al., 2024) proposed a dual-personalizing adapter (FedDPA), while (Qi et al., 2024) introduced FDLoRA. Both methods follow a similar approach by equipping each client with a personalized LoRA module and a global LoRA module, enabling the capture of both client-specific and global knowledge.

Another stream of research explores heterogeneous LoRA. (Cho et al., 2024) propose a framework that assigns different LoRA ranks to individual clients, then aggregates these heterogeneous modules via zero-padding, followed by redistribution through truncation. However, this straightforward zero-padding approach can lead to instability during training, as noted by (Byun & Lee, 2025). To address this challenge, (Byun & Lee, 2025) introduce a replication-based aggregation technique tailored for rank-heterogeneous LoRA. (Qiu et al., 2024) develop Rank-Based LoRA Aggregation (RBLA), which employs a weighted scheme to aggregate heterogeneous LoRA modules. (Wang et al., 2024) further contribute by presenting a stacking-based method to integrate diverse LoRA structures.

Notably, the aggregation errors introduced by applying LoRA in federated learning settings is under attention recently. (Sun et al., 2024) proposes FFA-LoRA, which fixes the randomly initialized non-zero $A$ matrices and only updates and aggregates the zero-initialized $B$ matrices to fix the aggregation errors and further halve the communication cost. However, because certain matrices remain fixed, LoRA's capacity to adapt is limited, which often leading to suboptimal performance (Guo et al., 2025; Zhang et al., 2023). FedSA-LoRA (Guo et al., 2025) claims that $A$ matrices are used to learn general knowledge while $B$ matrices focus on modeling client-specific knowledge. Based on that, they keep both $A$ and $B$ matrices trainable, but only share the $A$ matrices with the server for aggregation, and keep $B$ matrices personalized to each client. As a result, this approach faces the same limitation as other personalized FL methods: the global model aggregated at the central server is not truly general. FLoRA (Wang et al., 2024) instead stacks low-rank updates from all clients and redistributes them, but this breaks FedAvg convergence guarantees and leads to heavy communication overhead that grows exponentially with the number of clients in each round. FedEx-LoRA (Singhal

et al., 2025) computes the aggregation error on the server, sends the error weights to the clients, and updates the local model by the weights while preserving the aggregated LoRA. This cause extreme communication overhead, which totally eliminates the lightweight advantage of LoRA. ECLoRA (Ning et al., 2025) leverages randomized SVD to substantially reduce aggregation overhead, while integrating an error compensation mechanism that accounts for decomposition errors accumulated from previous rounds, thereby enhancing the precision of the aggregation process.

## C  EXPERIMENTAL DETAILS

### C.1  DATASETS

We evaluate FLoRA-NA across three computational prediction tasks: (1) language understanding, (2) mathematical reasoning, and (3) Code-solving ability.

**Language understanding.**   We use seven tasks from the GLUE benchmark (Wang et al., 2018), a widely adopted suite for evaluating natural language understanding:

- MNLI (Multi-Genre Natural Language Inference): requires determining whether a hypothesis sentence entails, contradicts, or is neutral with respect to a premise, across multiple genres.
- SST-2 (Stanford Sentiment Treebank): a binary classification task to predict the sentiment (positive/negative) of movie reviews.
- MRPC (Microsoft Research Paraphrase Corpus): evaluates whether two sentences are semantically equivalent (paraphrase detection).
- QNLI (Question Natural Language Inference): decide whether a context sentence contains the answer to a given question.
- QQP (Quora Question Pairs): detects whether two Quora questions are semantically equivalent.
- RTE (Recognizing Textual Entailment): a binary entailment classification task, built from multiple RTE challenges.
- STS-B (Semantic Textual Similarity Benchmark): predicts a similarity score (from 1 to 5) for a pair of sentences.

**Mathematical reasoning.**   We fine-tune the models on the MetaMathQA dataset (Yu et al., 2024) to enhance their mathematical problem-solving capabilities, and evaluate performance on GSM8K (Cobbe et al., 2021) and MATH (Yu et al., 2024).

**Code-solving ability.**   We fine-tune the models on the CodeFeedback dataset (Zheng et al., 2024) and evaluated using HumanEval (Chen et al., 2021) and MBPP (Austin et al., 2021).

### C.2  BASELINE METHODS.

We compare our method against several recent advances in federated fine-tuning with LoRA:

- FedIT (Zhang et al., 2024a): A straightforward baseline that directly integrates LoRA into federated learning. In each communication round, local clients fine-tune LoRA modules and the server aggregates them by averaging the $A$ and $B$ matrices separately. Despite its simplicity, FedIT often struggles with data heterogeneity and noise sensitivity.
- FedDPA-LoRA (Yang et al., 2024): A dual-personalizing adapter framework designed to address test-time distribution shifts in federated foundation models. It introduces both a global adapter (for generalization) and a local adapter (for personalization), which are combined via an instance-wise dynamic weighting mechanism during inference. This method partially addresses the generalization problem in the global model. However, introducing and training two separate adapters doubles the computational overhead and leads to additional latency, making it unsuitable for resource-constrained systems.
- FFA-LoRA (Sun et al., 2024): A communication-efficient and stable variant of LoRA for federated learning. FFA-LoRA freezes the randomly initialized non-zero matrices and only fine-tunes the zero-initialized matrices. This reduces the instability caused by data heterogeneity, differential privacy noise, and hyperparameter sensitivity, while halving communication costs compared to vanilla applying LoRA to FL. However, this approach effectively halves the expressive power of LoRA and transmits less information to the server, which ultimately leads to degraded performance.

- FedSA-LoRA (Guo et al., 2025): A method motivated by the asymmetry analysis of LoRA matrices. FedSA-LoRA observes that $A$ matrices capture general knowledge, whereas $B$ matrices encode client-specific information. Hence, only the $A$ matrices are shared with the server for aggregation, while $B$ matrices remain local. This strategy improves personalization, reduces communication overhead. However, this methods is designed for personalized scenario, and can not produce a general global model.

- FLoRA (Wang et al., 2024): An aggregation-noise-free method for federated fine-tuning that supports heterogeneous LoRA modules. Instead of averaging, FLoRA stacks the local LoRA matrices from different clients to construct the global modules. This method introduces additional communication overhead, growing linearly with the number of participating clients per communication round.

- FedEx-LoRA (Singhal et al., 2025) calculates a residual error matrix to the frozen pertained matrix on the server and then send it to each client to correct the error bring by the LoRA matrix aggregation on the server. Due to the enormous size of residual error matrix, this method is unbearable in practical FL system with limited bandwidth.

## C.3 ARCHITECTURE DETAILS.

**Language understanding**   We adopt the RoBERTa-large model (355M) (Liu et al., 2019) from the HuggingFace Transformers library (Wolf et al., 2020) as the base model, following (Guo et al., 2025).

**Mathematical reasoning and Code-solving ability**   We evaluate using large-scale decoder-based LLMs, including LLaMA 2-7B (Touvron et al., 2023), Mistral-7B-v0.1 (Jiang et al., 2023), and Gemma-7B (Team et al., 2024). This is our effort to provide a comprehensive overview and demonstrate the effectiveness of FLoRA-NA across various model architecture.

## C.4 EVALUATIONS METRICS.

To assess the global-local generalization, we evaluate the global aggregated model and the local personalized model on the global and local test datasets, respectively. However, for single-matrix aggregation methods, where only one LoRA matrix is transmitted to the server, direct evaluation of the global model is not feasible because the server lacks the complete parameter set. To address this issue, we approximate the global evaluation by using the local model at epoch 0, which is re-initialized with the local trained matrix combined with the global aggregated parameters. This metric will guarantee the fair comparisons among different FedLoRA algorithms. For instance, the global and local accuracy are measured as follows:

$$\text{Acc}_{\text{global}} = \text{Acc}(\mathcal{D}^{\text{test}}; W_u^{(i,0)}) = \mathbb{E}_{(x,y) \in \mathcal{D}^{\text{test}}}[\mathbb{1}(f(x; W_u^{(i,0)}) = y)], \tag{70}$$

$$\text{Acc}_{\text{u}} = \text{Acc}(\mathcal{D}_u^{\text{test}}; W_u^{(i,E)}) = \mathbb{E}_{(x,y) \in \mathcal{D}_u^{\text{test}}}[\mathbb{1}(f(x; W_u^{(i,E)}) = y)], \tag{71}$$

where $\mathbb{1}$ is the indicator function. Then, the generalization gap is measured as follows:

$$\text{Gen-Gap} = \frac{1}{U} \sum_{u=1}^{U} \text{Acc}_{\text{u}} - \text{Acc}_{\text{global}}. \tag{72}$$

## C.5 IMPLEMENTATION DETAILS.

For all tasks, including language understanding, mathematical reasoning, and code-solving, we consider a federated learning system with 10 clients. To ensure fair comparison with prior work, we adopt the same settings as FFA-LoRA (Sun et al., 2024) and FedSA-LoRA (Guo et al., 2025). The datasets are randomly partitioned among clients under a non-IID setting, using a Dirichlet distribution with parameter $\alpha = 0.5$ (Dir(0.5)).

For optimization, we employ different strategies: the SGD optimizer is used to update low-rank matrices, while the AdamW optimizer is applied to nearly accurate mechanism in case of FLoRA-NA, which ensures more accurate aggregation. Training is carried out with a batch size of 128, 10 local update steps per round, and 1000 total communication rounds, keeping the configuration consistent across all experiments.

Regarding LoRA insertion:

- Language understanding tasks: Following (Guo et al., 2025; Hu et al., 2022), we only apply LoRA to the $W_q$ and $W_v$ in the attention layers, with a dropout rate of 0.05.

- Mathematical reasoning and code-solving tasks: we apply LoRA to the $W_q$, $W_v$, $W_o$ in the attention layers and gating modules, with a dropout rate of 0.1.

In all cases, the rank is fixed at $r = 8$ and the scaling factor at $\alpha = 16$, following (Hu et al., 2022). For all tasks, the learning rate for each method is grid-searched over $\{1e-4, 2e-4, 1e-3, 2e-3, 5e-3, 1e-2, 2e-2, 5e-2\}$ (see Table 7 and Table 8). We observe that HiRA requires relatively larger learning rates to converge. Also, FLoRA-NA consistently supports higher learning rates compared to other approaches.

Table 7: Grid-searched learning rates for language understanding task in GLUE benchmark datasets.

| Variant | Method | MNLI | SST-2 | MRPC | QNLI | QQP | RTE | STS-B |
|---|---|---|---|---|---|---|---|---|
| LoRA (ICLR'22) | FedIT-LoRA | 2e-4 | 2e-4 | 1e-4 | 1e-4 | 5e-4 | 2e-4 | 2e-4 |
| | FedDPA-LoRA | 2e-4 | 1e-4 | 1e-4 | 5e-4 | 1e-4 | 2e-4 | 2e-4 |
| | FFA-LoRA | 5e-4 | 1e-4 | 1e-4 | 2e-4 | 5e-4 | 2e-4 | 2e-4 |
| | FedSA-LoRA | 5e-4 | 1e-3 | 2e-4 | 5e-4 | 2e-4 | 2e-4 | 5e-4 |
| | FLoRA | 1e-3 | 2e-4 | 1e-3 | 2e-3 | 2e-3 | 1e-3 | 5e-4 |
| | FedEx-LoRA | 1e-3 | 2e-4 | 1e-3 | 2e-3 | 2e-3 | 5e-4 | 5e-4 |
| | FLoRA-NA | 2e-3 | 5e-4 | 2e-3 | 1e-3 | 1e-3 | 5e-4 | 1e-3 |
| DoRA (ICML'24) | FedIT-DoRA | 5e-3 | 2e-3 | 1e-3 | 2e-3 | 2e-3 | 5e-4 | 1e-3 |
| | FedDPA-DoRA | 5e-4 | 2e-4 | 5e-4 | 1e-3 | 1e-3 | 2e-4 | 5e-4 |
| | FFA-DoRA | 1e-3 | 5e-4 | 1e-3 | 2e-3 | 5e-4 | 2e-4 | 5e-4 |
| | FedSA-DoRA | 2e-3 | 1e-3 | 2e-3 | 2e-3 | 1e-3 | 2e-4 | 5e-4 |
| | FDoRA | 2e-3 | 1e-3 | 2e-3 | 5e-3 | 2e-4 | 5e-4 | 1e-3 |
| | FedEx-DoRA | 2e-3 | 2e-4 | 5e-4 | 1e-3 | 2e-3 | 2e-3 | 1e-3 |
| | FDoRA-NA | 1e-2 | 2e-3 | 5e-3 | 5e-3 | 5e-3 | 2e-3 | 5e-3 |
| HiRA (ICLR'25) | FedIT-HiRA | 5e-3 | 2e-3 | 5e-3 | 2e-3 | 1e-3 | 1e-3 | 5e-3 |
| | FedDPA-HiRA | 5e-3 | 2e-3 | 1e-2 | 2e-3 | 5e-3 | 5e-4 | 4e-3 |
| | FFA-HiRA | 5e-3 | 2e-3 | 1e-3 | 2e-3 | 1e-3 | 5e-3 | 2e-3 |
| | FedSA-HiRA | 1e-2 | 5e-3 | 5e-3 | 2e-3 | 2e-3 | 1e-3 | 5e-3 |
| | FHiRA | 2e-3 | 1e-3 | 1e-2 | 5e-3 | 5e-3 | 5e-3 | 1e-2 |
| | FedEx-HiRA | 5e-3 | 2e-3 | 2e-3 | 5e-3 | 1e-2 | 1e-2 | 5e-3 |
| | FHiRA-NA | 2e-2 | 1e-2 | 2e-2 | 2e-2 | 1e-2 | 2e-2 | 1e-2 |

For validation on mathematical reasoning and code-solving ability, the learning rate is found the same way and shown in table 8. It can be seen that FLoRA-NA consistently supports higher learning rates compared to other approaches.

Table 8: Grid-searched learning rates used for different methods across datasets on mathematical reasoning and code-solving ability.

| Model | Method | GSM8K | MATH | HumanEval | MBPP |
|---|---|---|---|---|---|
| LLaMA-2-7B | FedIT-LoRA | 2e-4 | 1e-4 | 1e-4 | 2e-4 |
| | FFA-LoRA | 1e-4 | 2e-4 | 5e-4 | 5e-4 |
| | FedSA-LoRA | 5e-4 | 2e-4 | 2e-4 | 2e-4 |
| | FLoRA | 1e-3 | 5e-4 | 5e-4 | 2e-4 |
| | FedEx-LoRA | 1e-3 | 5e-4 | e-4 | 1e-5 |
| | FLoRA-NA | 2e-3 | 1e-3 | 2e-3 | 1e-3 |
| Mistral-7B | FedIT-DoRA | 5e-4 | 1e-4 | 2e-4 | 1e-4 |
| | FFA-LoRA | 5e-4 | 1e-4 | 1e-4 | 2e-4 |
| | FedSA-LoRA | 1e-3 | 2e-4 | 1e-4 | 5e-4 |
| | FLoRA | 1e-3 | 5e-4 | 2e-4 | 1e-3 |
| | FedEx-LoRA | 2e-3 | 5e-4 | 2e-4 | 5e-4 |
| | FLoRA-NA | 5e-3 | 5e-4 | 1e-3 | 2e-3 |
| Gemma-7B | FedIT-HiRA | 2e-4 | 1e-4 | 2e-4 | 5e-4 |
| | FFA-LoRA | 2e-4 | 5e-4 | 1e-4 | 1e-4 |
| | FedSA-LoRA | 1e-3 | 2e-4 | 5e-4 | 1e-4 |
| | FLoRA | 2e-3 | 1e-3 | 5e-4 | 2e-4 |
| | FedEx-LoRA | 1e-3 | 5e-4 | 5e-4 | 2e-4 |
| | FLoRA-NA | 5e-3 | 1e-3 | 2e-3 | 1e-3 |

# D ADDITIONAL RESULTS

## D.1 STANDARD DEVIATIONS

From table 9, we observe that in the results on local validation are similar between method and significantly smaller than the case of global validation, which is reasonable. Specifically, when validating on global dataset, we observe that all baseline variants of LoRA, DoRA, and HiRA exhibit relatively high fluctuations (often exceeding 2.5 and up to nearly 4), reflecting instability under non-IID client distributions. In contrast, our proposed neraly accurate aggregation variants (FLoRA-NA, FDoRA-NA, and FHiRA-NA) consistently achieve substantially lower variance across all tasks, with deviations generally below 2.2. This demonstrates that incorporating noise-awareness not only stabilizes training but also improves robustness against randomness in federated optimization, enabling more reliable deployment in practical FL scenarios. Similar trend is observed in table 10.

Table 9: Standard deviation on GLUE benchmark datasets across 5 seeds. The results are averaged across all clients.

| Variant | Method | MNLI | SST-2 | MRPC | QNLI | QQP | RTE | STS-B | Avg |
|---|---|---|---|---|---|---|---|---|---|
| LoRA (ICLR'22) | FedIT-LoRA | 1.32 \| 2.92 | 1.47 \| 2.57 | 1.28 \| 3.38 | 1.55 \| 3.15 | 1.82 \| 2.92 | 1.34 \| 2.94 | 1.61 \| 2.71 | 1.48 \| 2.94 |
| | FedDPA-LoRA | 1.53 \| 2.63 | 1.15 \| 3.25 | 1.81 \| 3.91 | 1.38 \| 2.48 | 1.62 \| 2.72 | 1.26 \| 3.36 | 1.85 \| 2.95 | 1.51 \| 3.04 |
| | FFA-LoRA | 1.75 \| 2.85 | 1.33 \| 2.43 | 1.46 \| 3.16 | 1.87 \| 3.97 | 1.54 \| 2.64 | 1.29 \| 3.39 | 1.72 \| 2.82 | 1.57 \| 3.04 |
| | FedSA-LoRA | 1.12 \| 2.92 | 1.64 \| 3.74 | 1.48 \| 2.58 | 1.73 \| 2.83 | 1.36 \| 2.46 | 1.59 \| 2.69 | 1.77 \| 3.87 | 1.53 \| 3.01 |
| | FLoRA | 1.54 \| 2.86 | 1.32 \| 3.46 | 1.44 \| 2.35 | 1.65 \| 2.12 | 1.39 \| 2.34 | 1.46 \| 2.15 | 1.83 \| 3.35 | 1.52 \| 2.66 |
| | FedEx-LoRA | 1.12 \| 2.92 | 1.64 \| 3.74 | 1.58 \| 2.18 | 1.64 \| 2.13 | 1.39 \| 2.26 | 1.94 \| 2.19 | 1.45 \| 2.47 | 1.54 \| 2.21 |
| | FLoRA-NA | 1.35 \| 2.15 | 1.72 \| 1.92 | 1.59 \| 2.29 | 1.51 \| 1.81 | 1.87 \| 2.47 | 1.48 \| 1.98 | 1.32 \| 1.62 | 1.55 \| 2.03 |
| DoRA (ICML'24) | FedIT-DoRA | 1.26 \| 3.36 | 1.39 \| 2.49 | 1.17 \| 3.27 | 1.48 \| 2.58 | 1.63 \| 2.73 | 1.24 \| 3.34 | 1.59 \| 2.69 | 1.39 \| 2.92 |
| | FedDPA-DoRA | 1.48 \| 2.58 | 1.22 \| 3.32 | 1.77 \| 2.87 | 1.32 \| 3.42 | 1.58 \| 2.68 | 1.19 \| 3.29 | 1.80 \| 2.90 | 1.48 \| 3.01 |
| | FFA-DoRA | 1.64 \| 2.74 | 1.41 \| 3.51 | 1.33 \| 3.43 | 1.72 \| 2.82 | 1.59 \| 2.69 | 1.31 \| 3.41 | 1.66 \| 3.76 | 1.52 \| 3.19 |
| | FedSA-DoRA | 1.76 \| 2.86 | 1.52 \| 2.62 | 1.45 \| 3.55 | 1.67 \| 2.77 | 1.39 \| 3.49 | 1.55 \| 2.65 | 1.73 \| 2.83 | 1.58 \| 2.97 |
| | FDoRA | 1.62 \| 2.50 | 1.89 \| 2.13 | 1.76 \| 1.95 | 1.67 \| 2.35 | 1.71 \| 2.58 | 1.95 \| 2.85 | 1.71 \| 2.62 | 1.76 \| 2.43 |
| | FedEx-DoRA | 1.64 \| 2.42 | 1.51 \| 2.33 | 1.34 \| 1.94 | 1.57 \| 1.86 | 1.39 \| 2.11 | 1.62 \| 2.63 | 1.48 \| 1.84 | 1.51 \| 2.18 |
| | FDoRA-NA | 1.43 \| 2.13 | 1.51 \| 2.01 | 1.34 \| 1.74 | 1.57 \| 1.97 | 1.39 \| 2.29 | 1.62 \| 2.72 | 1.48 \| 1.88 | 1.48 \| 2.11 |
| HiRA (ICLR'25) | FedIT-HiRA | 1.19 \| 3.29 | 1.31 \| 2.41 | 1.08 \| 3.18 | 1.42 \| 2.52 | 1.54 \| 2.64 | 1.16 \| 3.26 | 1.45 \| 3.55 | 1.31 \| 2.98 |
| | FedDPA-HiRA | 1.44 \| 2.54 | 1.18 \| 3.28 | 1.69 \| 2.79 | 1.27 \| 2.37 | 1.52 \| 2.62 | 1.14 \| 3.24 | 1.76 \| 2.86 | 1.43 \| 2.81 |
| | FFA-HiRA | 1.72 \| 2.82 | 1.36 \| 2.46 | 1.29 \| 3.39 | 1.67 \| 2.77 | 1.51 \| 2.61 | 1.27 \| 3.37 | 1.63 \| 2.73 | 1.49 \| 2.88 |
| | FedSA-HiRA | 1.61 \| 2.71 | 1.48 \| 2.58 | 1.34 \| 3.44 | 1.59 \| 2.69 | 1.33 \| 3.49 | 1.55 \| 3.61 | 1.69 \| 2.79 | 1.51 \| 3.04 |
| | FHiRA | 1.52 \| 2.68 | 1.48 \| 2.75 | 1.41 \| 1.97 | 1.75 \| 2.62 | 1.81 \| 2.43 | 1.56 \| 2.72 | 1.62 \| 2.32 | 1.59 \| 2.50 |
| | FedEx-HiRA | 1.42 \| 2.62 | 1.75 \| 2.85 | 1.53 \| 1.97 | 1.61 \| 2.54 | 1.52 \| 2.36 | 1.75 \| 2.66 | 1.62 \| 2.15 | 1.60 \| 2.45 |
| | FHiRA-NA | 1.38 \| 2.48 | 1.45 \| 2.55 | 1.31 \| 1.81 | 1.53 \| 2.13 | 1.36 \| 2.06 | 1.49 \| 2.49 | 1.42 \| 2.02 | 1.42 \| 2.22 |

Table 10: Standard deviation on mathematical reasoning tasks (Cobbe et al., 2021) and code-solving ability tasks across 5 seeds using various backbone models.

| Model | Method | GSM8K | MATH | HumanEval | MBPP | Avg |
|---|---|---|---|---|---|---|
| LLaMA-2-7B | FedIT | 1.84 \| 2.73 | 1.55 \| 3.01 | 1.62 \| 2.67 | 1.76 \| 2.95 | 1.69 \| 2.84 |
| | FFA-LoRA | 1.76 \| 2.41 | 1.88 \| 2.64 | 1.64 \| 2.58 | 1.75 \| 2.70 | 1.76 \| 2.58 |
| | FedSA-LoRA | 1.61 \| 3.12 | 1.37 \| 3.20 | 1.75 \| 2.92 | 1.54 \| 3.08 | 1.57 \| 3.08 |
| | FLoRA | 1.73 \| 2.48 | 1.87 \| 2.63 | 1.91 \| 2.34 | 1.66 \| 2.68 | 1.79 \| 2.53 |
| | FedEx-LoRA | 1.60 \| 2.18 | 2.03 \| 2.37 | 1.81 \| 1.96 | 1.52 \| 2.73 | 1.74 \| 2.31 |
| | FLoRA-NA | 1.63 \| 2.08 | 1.97 \| 2.29 | 1.83 \| 1.91 | 1.66 \| 2.52 | 1.77 \| 2.20 |
| Mistral-7B | FedIT | 1.65 \| 2.88 | 1.77 \| 3.10 | 1.84 \| 2.77 | 1.92 \| 2.96 | 1.79 \| 2.93 |
| | FFA-LoRA | 1.72 \| 2.44 | 1.95 \| 2.68 | 1.86 \| 2.59 | 1.99 \| 2.71 | 1.88 \| 2.60 |
| | FedSA-LoRA | 1.78 \| 3.15 | 1.86 \| 3.20 | 1.67 \| 3.01 | 1.81 \| 3.18 | 1.78 \| 3.13 |
| | FLoRA | 1.76 \| 2.12 | 2.21 \| 2.89 | 1.81 \| 2.60 | 1.81 \| 2.23 | 1.86 \| 2.46 |
| | FedEx-LoRA | 1.65 \| 2.23 | 2.21 \| 2.73 | 1.77 \| 2.69 | 1.75 \| 2.27 | 1.84 \| 2.48 |
| | FLoRA-NA | 1.55 \| 1.93 | 2.11 \| 2.67 | 1.62 \| 2.19 | 1.83 \| 2.01 | 1.78 \| 2.20 |
| Gemma-7B | FedIT | 1.89 \| 2.74 | 1.54 \| 3.06 | 1.75 \| 2.82 | 1.92 \| 2.97 | 1.77 \| 2.90 |
| | FFA-LoRA | 1.68 \| 2.42 | 1.92 \| 2.69 | 1.79 \| 2.55 | 1.61 \| 2.64 | 1.75 \| 2.58 |
| | FedSA-LoRA | 1.51 \| 3.18 | 1.70 \| 3.22 | 1.51 \| 3.04 | 1.77 \| 3.20 | 1.62 \| 3.16 |
| | FLoRA | 1.52 \| 2.78 | 1.49 \| 2.53 | 1.47 \| 2.44 | 1.34 \| 2.86 | 1.46 \| 2.65 |
| | FedEx-LoRA | 1.57 \| 2.33 | 1.45 \| 2.03 | 1.64 \| 2.9 | 1.51 \| 2.71 | 1.54 \| 2.49 |
| | FLoRA-NA | 1.47 \| 2.15 | 1.39 \| 1.97 | 1.52 \| 2.16 | 1.44 \| 2.20 | 1.46 \| 2.12 |

## D.2 LAYER-WISE NORMALIZED FROBENIUS NORM OF DIVERGENCE BETWEEN IDEAL AND APPROXIMATE UPDATE.

To further demonstrate the robustness of FLoRA's nearly accurate estimation and highlight the differences between FLoRA and FedEx-LoRA, we compute and visualize the distance between the ideal and approximate updates across all layers. The visualization shows that FLoRA's nearly accurate optimization effectively minimizes these distances throughout the network. Moreover, the distances are reduced uniformly across all layers, in contrast to the results reported for FedEx-LoRA in (Singhal et al., 2025, Figure 2), where only the final layers achieve high precision.

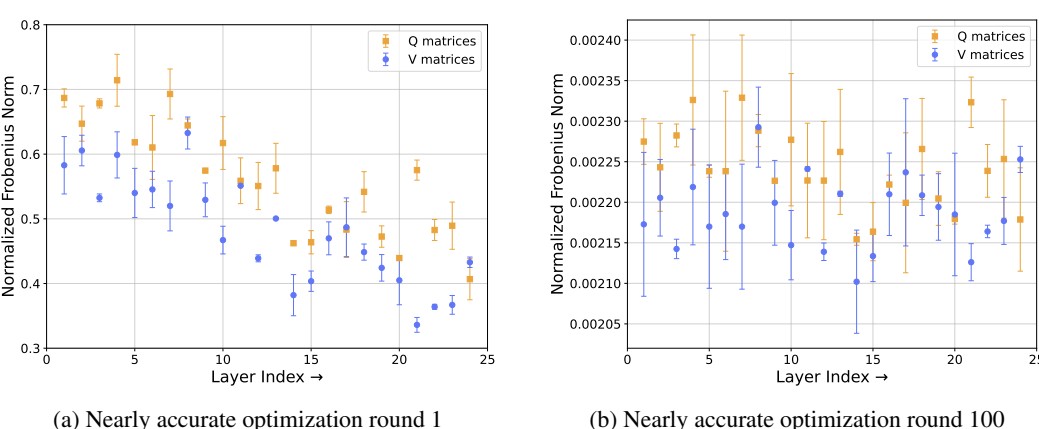

(a) Nearly accurate optimization round 1          (b) Nearly accurate optimization round 100

Figure 4: Comparison of the layer-wise normalized Frobenius norm of divergence between the gradient obtained from the ideal update and that from the approximate update under the naive FedAvg strategy with full-parameter, using FedIT and the proposed FLoRA-NA method. This experiment is conducted on MNLI dataset.

## D.3 NATURAL LANGUAGE GENERATING RESULTS ON GSM8K DATASET

Based on the provided accuracy scores, the table 11 effectively demonstrates the varying performance of the three methods on a mathematical reasoning task. The results clearly show that FLoRA-NA (42.89% accuracy) is the most effective method, as it not only provides the correct final answer but also presents a clear, well-structured, and mathematically sound solution. In contrast, FedIT (35.68% accuracy) performs at a moderate level, producing the correct answer but with a simpler, more direct explanation. The least effective method is FedSA-LoRA (28.82% accuracy), which fails to solve the problem correctly by making a critical error in the final calculation, highlighting its lower reliability. The differences in output quality directly correspond to the given accuracy scores, providing a concise and illustrative comparison of each method's capabilities.

Table 11: Natural Language Generating results on GSM8K Dataset using LLaMA-2-7B pretrained model.

| | FedIT | FedSA-LoRA | FLoRA-NA |
|---|---|---|---|
| Accuracy | 35.68 | 28.82 | **42.89** |
| Question: | Mr. Finnegan has 3 tanks with a capacity of 7000 gallons, 5000 gallons, and 3000 gallons, respectively. If he fills the first tank up to 3/4 full, the second tank with water up to 4/5 of its capacity, and the third tank up to half of its capacity, how many gallons in total are in the tanks? | | |
| Output: | To find the total volume, we must calculate the volume in each tank. Tank 1 has 3/4 of 7000 gallons, which is 5250 gallons. Tank 2 has 4/5 of 5000 gallons, which is 4000 gallons. Tank 3 has 1/2 of 3000 gallons, which is 1500 gallons. The total is 5250 + 4000 + 1500 = 10750 gallons. The answer is 10750. [✓] | Tank 1: 7000 gallons $\times$ 3/4 = 5250 gallons. The second tank is 4/5 of its capacity, so 5000 $\times$ 4/5 = 4000 gallons. The third tank is half full. That is 3000 $\times$ 1/2 = 1500. Total is 5250 + 4000 = 9250. The answer is 9250. [X] | Let $V_1, V_2, V_3$ be the capacities of the three tanks. We are given $V_1 = 7000$ gallons, $V_2 = 5000$ gallons, and $V_3 = 3000$ gallons. The filled volumes are $V_{1,\text{filled}} = \frac{3}{4}V_1 = \frac{3}{4} \times 7000 = 5250$ gallons. $V_{2,\text{filled}} = \frac{4}{5}V_2 = \frac{4}{5} \times 5000 = 4000$ gallons. $V_{3,\text{filled}} = \frac{1}{2}V_3 = \frac{1}{2} \times 3000 = 1500$ gallons. The total volume is $V_{\text{total}} = V_{1,\text{filled}} + V_{2,\text{filled}} + V_{3,\text{filled}} = 5250 + 4000 + 1500 = 10750$ gallons. The answer is 10750. [✓] |

## D.4 NATURAL LANGUAGE GENERATING RESULTS ON MBPP DATASET

We evaluate the natural language generation performance on the MBPP dataset using the LLaMA-2-7B pretrained model. The results show that FedIT achieves an accuracy of 28.37, FedSA-LoRA obtains 25.02, and FLoRA-NA achieves the highest score of 35.10, clearly demonstrating its superior capability. To better understand the performance differences among FedIT, FedSA-LoRA, and FLoRA-NA, we consider the following problem from the MBPP dataset:

> *Write a function to remove characters from the first string which are present in the second string.*

The following three test cases are provided in MBPP dataset:

```
assert remove_dirty_chars("probasscurve", "pros") == "bacuve"
assert remove_dirty_chars("digitalindia", "talent") == "digiidi"
assert remove_dirty_chars("exoticmiles", "toxic") == "emles"
```

**FedIT [X].** The solution generated by FedIT is shown below:

```
def remove_dirty_chars(str1, str2):
    temp_list = []
    index = 0
    while index < len(str1):
        char = str1[index]
        should_add = True
        for j in range(len(str2)):
            if j < len(str2):
                if char == str2[j]:
                    should_add = False
                    break
        if should_add == True:
            temp_list.append(char)
        else:
            temp_list.append("_")
        index += 1
    result = " ".join(temp_list)
    final = result.replace("_", "")
    return final
```

The outputs produced by this implementation are:

```
remove_dirty_chars("probasscurve", "pros")   -> b a   c u  v e
remove_dirty_chars("digitalindia", "talent") -> d i g i    i  d i
remove_dirty_chars("exoticmiles", "toxic")   -> e      m  l e s
```

Although partially correct in terms of character removal, the output contains extra spaces and formatting errors, leading to deviations from the expected results.

**FedSA-LoRA [X].** The solution generated by FedSA-LoRA is:

```
def remove_dirty_chars(s1, s2):
    s2_list = []
    for ch in s2:
        if ch not in s2_list:
            s2_list.append(ch)
    result = ""
    for ch in s1:
        remove = False
        for r in s2_list:
            if r < ch:
```

```
                    remove = True
        if not remove:
            result += ch
    return result
```

The outputs produced are:

```
remove_dirty_chars("probasscurve", "pros")    -> obace
remove_dirty_chars("digitalindia", "talent") -> aa
remove_dirty_chars("exoticmiles", "toxic")    -> c
```

Here the logic is incorrect because of an erroneous use of relational ordering instead of equality. As a result, the outputs significantly deviate from the expected ones.

**FLoRA-NA [✓].** The solution generated by FLoRA-NA is:

```
def remove_dirty_chars(str1, str2):
    n = len(str1)
    i = 0
    result_chars = []
    chars_to_remove = set(str2)
    while i < n:
        current_char = str1[i]
        if current_char not in chars_to_remove:
            result_chars.append(current_char)
        i += 1
    final_result = "".join(result_chars)
    return final_result
```

The outputs are:

```
remove_dirty_chars("probasscurve", "pros")    -> bacuve
remove_dirty_chars("digitalindia", "talent") -> digiidi
remove_dirty_chars("exoticmiles", "toxic")    -> emles
```

This implementation correctly removes the characters and produces exactly the expected results for all test cases.

From this case study, it is evident that while FedIT and FedSA-LoRA fail to consistently solve the task, FLoRA-NA provides a concise and robust solution that generalizes correctly across all inputs. Combined with its highest quantitative accuracy (35.10) on the MBPP benchmark, this strongly confirms the superiority of FLoRA-NA for natural language generation tasks.

# E   DISCUSSION ON PRIVACY/SECURITY

Numerous attacks are prevalent in FL, both in general and specifically in the context of FedAvg (McMahan et al., 2017). These include data poisoning, model poisoning , backdoor attacks (Nguyen et al., 2023; Hung-Quang et al., 2025), and gradient inversion attacks (Dimitrov et al., 2024). Our proposed approach does not introduce any additional privacy risks beyond those already present in FedAvg. Consequently, it remains fully compatible with existing defense mechanisms designed for FedAvg, such as secure aggregation (Nguyen et al., 2022) or noise injection (Phan et al., 2025) prior to aggregation.

In contrast to several existing methods (Singhal et al., 2025; Wang et al., 2024) where the server must broadcast extensive residual parameters $\nabla \hat{W}$, or stacking LoRA matrices in addition to the LoRA matrices $\hat{A}$ and $\hat{B}$, FLoRA-NA only requires transmitting the LoRA matrices $\hat{A}$ and $\hat{B}$. This design choice minimizes communication overhead and avoids introducing further privacy concerns. Moreover, unlike FedAvg, where the aggregation rule is simply parameter averaging, the aggregation process in FLoRA-NA is not explicitly known to clients or potential adversaries. This obscurity increases the difficulty for adversaries attempting to perform gradient inversion attacks, thereby enhancing the privacy and security of the federated LoRA system.

