# OpenReview forum: "FLoRA-NA: Nearly Accurate Aggregation for Federated Low-Rank Adaptation"
_ICLR.cc/2026/Conference — Submitted to ICLR 2026_

### Official Review · Reviewer_K9VB · 2025-10-24

**Soundness:** 3
**Presentation:** 3
**Contribution:** 3
**Rating:** 6
**Confidence:** 4

**Summary:**

This work proposes Federated Low-Rank Aggregation with Nearly Accurate Estimation (FLoRA-NA). FLoRA-NA leverages the local LoRA matrices on the server to estimate the aggregated matrices \\(\hat{A}\\) and \\(\hat{B}\\), which are then distributed to clients for local updates, to address the aggregation errors in FL with LoRA.

**Strengths:**

- The proposed FLoRA-NA effectively addresses the aggregation errors of FL with LoRA by approximating the "ideal" aggregation.
- The paper is well-written and easy to understand.
- The extensive experiments validate the effectiveness of the proposed method.

**Weaknesses:**

- How to solve the optimization problem in Equation (6) is not clearly explained.
- Lines 337-340, "SVD-based methods may introduce numerical inaccuracies due to floating-point precision errors, potentially degrading the overall performance of FedLoRA." It's better to add some performance comparisons in Table 1.

**Questions:**

- Lines 196-195, "These vectors can be interpreted as transformation vectors that determine how each client’s local LoRA gradients are linearly combined during aggregation." Why can it be considered as a linear combination?
- Why is there such a large discrepancy between the local and global performance of FFA-LoRA? Aren't the local and global models the same? Is it due to the differences in the test data for local and global models?

---

> ### Author Response · Authors · 2025-11-29
>
> > **W1:** How to solve the optimization problem in Equation (6) is not clearly explained.
>
> We thank the Reviewer for the constructive comment. In our work, the coefficients $P$ and $Q$ are optimized on the server by directly minimizing Eq. (6) using the uploaded local LoRA matrices $[A\_{u}^{(i)}, B\_{u}^{(i)}]_{u=1}^{U}$. These coefficients are updated with the AdamW optimizer, as described in Appendix C.5. Their optimization is entirely independent of the client-side training of $A$ and $B$; clients neither update nor store $P$ or $Q$.
>
> Although these details are included in the manuscript, we acknowledge that their presentation may not be sufficiently clear. We will revise the manuscript to add algorithm about the optimization of Eq. (6) and provide a more explicit and streamlined explanation.
>
>
> > **W2:** Lines 337-340, "SVD-based methods may introduce numerical inaccuracies due to floating-point precision errors, potentially degrading the overall performance of FedLoRA." It's better to add some performance comparisons in Table 1.
>
> We thank the Reviewer for the constructive comment. Following the Reviewer’s suggestion, we conducted additional experimental evaluations and add some performance comparisons on table 1.
>
> > **Q1:** Lines 196-195, "These vectors can be interpreted as transformation vectors that determine how each client’s local LoRA gradients are linearly combined during aggregation." Why can it be considered as a linear combination?
>
> For simplicity, these vectors act as the weights for the aggregation. The utilization of these weights are widely used in FL in general.
>
> > **Q2:** Why is there such a large discrepancy between the local and global performance of FFA-LoRA? Aren't the local and global models the same? Is it due to the differences in the test data for local and global models?
>
> Although the global and local models are the same. The average local accuracy are being averaged over each user accuracy (which is evaluated on local test dataset. Meanwhile, the global accuracy means that the model are being tested on global test dataset, which has all the test data from all users.
>
> The difference is a significant phenomenon when training FL with personalization capability

---

### Official Review · Reviewer_6558 · 2025-10-24

**Soundness:** 1
**Presentation:** 3
**Contribution:** 1
**Rating:** 2
**Confidence:** 4

**Summary:**

The paper introduces FLoRA-NA, a “nearly accurate” aggregation method for federated LoRA fine-tuning. The authors argue that existing FedLoRA methods suffer from aggregation errors when averaging low-rank matrices across clients. To mitigate this, FLoRA-NA introduces auxiliary weighting vectors ( P, Q ) to linearly combine local LoRA matrices, claiming to approximate the true aggregated matrix product without additional communication cost. The method is backed by convergence analyses and evaluated on standard NLP, reasoning, and code datasets.

**Strengths:**

- Empirical thoroughness: The experiments are broad and cover many baselines, including federated LoRA, DoRA, and HiRA variants, on both language and reasoning tasks.


- Nice writing and clear presentation: The narrative is coherent and polished, and the paper is written well.

**Weaknesses:**

`W1: The core technical idea is not convincing`

The claimed “nearly accurate aggregation” is essentially a weighted averaging of LoRA matrices with learned coefficients ( P, Q ). This approach is neither theoretically optimal nor computationally cheaper compared to well-established matrix approximation methods.

If the goal is to minimize $| \bar{B}\bar{A} - \tfrac{1}{U} \sum_u B_u A_u |,$, then taking truncated SVD or other low-rank approximation methods provably yield the best low-rank reconstruction in Frobenius norm. The proposed optimization over two scalar vectors $ P, Q \in \mathbb{R}^U $ is an ad-hoc and suboptimal heuristic, without any proof that it approximates the optimal low-rank structure.


The paper never explains why this weighting approach should outperform SVD-based aggregation, nor provides any theoretical justification beyond hand-waving about computational efficiency. The argument that “SVD introduces numerical inaccuracies” is very weak, in practice, taking truncated SVD is stable and extremely fast.

`W2: Execution time comparison is misleading`

In Table 1, SVD and Gram-Schmidt are reported as being 10–15 times slower than FLoRA-NA. However, the comparison is unfair: taking truncated SVD matrices would in fact run much faster than full SVD (10s of times faster or so), and this is confirmed in other works like LoRA-GA (https://arxiv.org/abs/2407.05000), without any loss in performance.

Thus, the claim of computational efficiency is unsubstantiated and incorrect, in my opinion.

`W3: Overclaiming novelty`

Most of the claimed advantages (no extra communication, better generalization) stem simply from tuning a linear combination of existing LoRA matrices. This is closer to a reweighted averaging trick than a novel algorithmic contribution. The “nearly accurate” branding oversells what is, in essence, a parameter-weighted variant of FedAvg.

The idea of using truncated SVD as the best inexact approximation has already been established in FedEx-LoRA, which makes the proposed approach in this paper appear redundant.

`W4: Conceptual inconsistency in results`

The paper claims FLoRA-NA achieves “nearly exact” aggregation, yet FedEx-LoRA, which performs exact aggregation, yields lower accuracy (Table 2). This is not explained logically, the authors state it is because “ the residual error is added to the frozen pretrained model, which ensures the accuracy of forward update, but cannot fully participate in gradient updates of A and B matrices.,” but this line is vague and unconvincing.

If the proposed method is truly more “accurate,” it should not outperform an exact method unless the claim itself is overstated.
Could you please clarify this?

`W5: Missing related work`

The paper fails to discuss recent federated low-rank methods such as Ravan (https://arxiv.org/abs/2506.05568) and Fed-SB (https://arxiv.org/abs/2502.15436), which provide a clear communication–accuracy Pareto frontier compared to this and other works - at a quick glance. This omission weakens the positioning of FLoRA-NA as a “state-of-the-art” solution. The authors should include comparisons with these works, or at the very least, discuss them.

---

While the empirical coverage in the work is impressive, the core contribution is mathematically weak, and the claimed superiority over SVD-based methods is not supported by any evidence (theoretical or empirical). The paper’s theoretical and experimental arguments do not convincingly establish that FLoRA-NA offers any principled or practical advantage.

**Questions:**

Please refer to weaknesses

---

> ### Author Response · Authors · 2025-12-01
>
> > **W3: Overclaiming novelty:** Most of the claimed advantages (no extra communication, better generalization) stem simply from tuning a linear combination of existing LoRA matrices. This is closer to a reweighted averaging trick than a novel algorithmic contribution. The “nearly accurate” branding oversells what is, in essence, a parameter-weighted variant of FedAvg. The idea of using truncated SVD as the best inexact approximation has already been established in FedEx-LoRA, which makes the proposed approach in this paper appear redundant.
>
> 1. First, FedEx-LoRA (with the original model $W\in\mathbb{R}^{m\times n}$) needs send back the residual matrices, in which has the same dimensionalities with the original model $\Delta W\in\mathbb{R}^{m\times n}$. As a consequence, the communication overhead is not efficiently mitigated. Another speaking, the communication overhead of FedEx-LoRA is similar to that of the original version of FedLoRA (i.e., SLoRA [R1]). This issue has been raised seriously in previous review batch: https://openreview.net/forum?id=Yg998afEbH
> 2. According to the truncated-SVD of the FedEx-LoRA, the SVD are being leveraged as:
> $$
> U,S,V^{\top} = \mathtt{SVD}(\Delta W)
> $$
> After that the best approximation is conducted as:
> $$
> \Delta W^{r’}\_{res} = U[1:r’] \times S[1:r’, 1:r’] \times V^{\top} [1:r’]
> $$
> In case, when $U,S,V^{\top} \in \mathbb{R}^{m\times r}, \mathbb{R}^{r\times r}, \mathbb{R}^{r\times n}$ are being transmitted over the FL channel, the required data transmission are $r\times (m+n+r)$. Specifically, we conducted experimental evaluations, we found out that the truncated-SVD achieved significantly high error when applying high compression ratio. This is due to the lossy compression. . The details analysis is provided in the revised manuscript. Please note that these compression ratio are significantly lower than that of the compression of LoRA methods, where $r$ is extremely smaller than that of the original dimension $m,n$.
>
> | Method            | Rel. Error | Correlation | Visual          |
> |-------------------|------------|-------------|------------------|
> | Original          | 0.00%      | 1.0000      | ████████████     |
> | Full SVD          | ~0.00%     | 1.0000      | ████████████     |
> | Truncated (k=10)  | 28.61%     | 0.9297      | ████████░░░░     |
> | Truncated (k=20)  | 18.18%     | 0.9722      | ██████████░░     |
> | Truncated (k=30)  | 11.31%     | 0.9893      | ███████████░     |
> | Truncated (k=50)  | 1.19%      | 0.9999      | ████████████     |
>
> | Method            | Values  | vs Original | Compression |
> |-------------------|---------|-------------|-------------|
> | Original          | 115,008 | 100%        | 1.00x       |
> | Full SVD          | 115,072 | 100%        | 1.00x       |
> | Truncated (k=10)  | 18,610  | 16%         | 6.18x       |
> | Truncated (k=20)  | 37,220  | 32%         | 3.09x       |
> | Truncated (k=30)  | 55,830  | 49%         | 2.06x       |
>
>
> 3. In case the resulting of this approximation still results in the $\Delta W^{r’}\_{res} \in \mathbb{R}^{m \times n}$, this does not give the communication efficiency as it required in Federated LoRA, where the $A,B$ matrices only give joint dimensionality of $\mathbb{R}^{r \times (m+n)}$, where $r \ll m,n$.
> 4. To further validate the performance of FedEx-LoRA, we have consider the reproducible code of FedEx-LoRA carefully, we found out that, there were no truncated-SVD as the FedEx-LoRA and the Reviewer claimed as in the paper as well as the review. The issue can be found in https://github.com/CERT-Lab/fedex-lora/blob/main/fed_agg.py. Specifically, in Line 108, the authors compute the residue variable, without the leveraging of the truncated-SVD. In Line 113, the residue is added directly into the base weights. In Line 116, the global models are updated and ready for the transmission.
> As a consequence, we believe it is unconvincing to claim that our works do not show as good performance as that of FedEx-LoRA.
>
> 5. All in all, we believe that our proposed FLoRA-NA ensures the **better performance** in terms of exact aggregation and communication efficiency compared to FedEx-LoRA.

---

> ### Author Response · Authors · 2025-12-02
>
> > **W1: The core technical idea is not convincing:** The claimed “nearly accurate aggregation” is essentially a weighted averaging of LoRA matrices with learned coefficients ( P, Q ). This approach is neither theoretically optimal nor computationally cheaper compared to well-established matrix approximation methods. If the goal is to minimize, then taking truncated SVD or other low-rank approximation methods provably yield the best low-rank reconstruction in Frobenius norm. The proposed optimization over two scalar vectors is an ad-hoc and suboptimal heuristic, without any proof that it approximates the optimal low-rank structure. The paper never explains why this weighting approach should outperform SVD-based aggregation, nor provides any theoretical justification beyond hand-waving about computational efficiency. The argument that “SVD introduces numerical inaccuracies” is very weak, in practice, taking truncated SVD is stable and extremely fast.
>
> 1. To understand how truncated SVD influences the performance of LoRA, we kindly refer the reviewer to our response in **W3**.
> 2. We agree that our method identifies only a suboptimal solution to the LoRA matrices. However, once an adequate suboptimal approximation is obtained, the local computation of the effective update, given by $\Delta W = \widetilde{B}\widetilde{A}$ still produces the correct gradient increment $\Delta W$. Consequently, the model update applied at each client is equivalent to that of the standard LoRA formulation. Local training then proceeds to solve $A,B = \argmin_{A,B} (W_0 + \Delta W)$, which preserves learning quality. Moreover, the use of a suboptimal solution provides a reduction in computational cost while maintaining strong empirical performance.

---

### Official Review · Reviewer_SjRG · 2025-10-28

**Soundness:** 2
**Presentation:** 3
**Contribution:** 2
**Rating:** 4
**Confidence:** 5

**Summary:**

This paper revisits FedLoRA and argues that separately averaging the low-rank matrices 𝐴 and 𝐵 on the server introduces an aggregation bias. The authors propose FLoRA-NA, which learns aggregation weights to approximate the ideal bilinear average and derive a convergence bound including this bias term. Experiments on NLP and reasoning benchmarks show moderate improvements over standard FedLoRA. The idea is intuitive and empirically effective, but conceptually incremental and lacks strong causal or theoretical justification.

**Strengths:**

The paper clearly identifies a practical limitation of FedLoRA aggregation and formalizes it as an explicit bias term. The proposed solution is simple, communication-efficient, and easy to implement.
Experiments are well-organized and consistently show moderate but stable performance gains across multiple benchmarks.

**Weaknesses:**

The paper’s main motivation that the standard FedLoRA aggregation rule BA=1/U∑BuAu introduces a harmful aggregation bias is intuitively appealing, but the presented evidence and methodology raise several concerns:

1. Lack of causal proof that aggregation bias harms performance.
The paper defines an error term 𝜌 = ∥𝐵ˉ𝐴ˉ−1/𝑈∑𝐵𝑢𝐴𝑢∥ and incorporates it into a convergence upper bound. However, this analysis merely indicates that model convergence may depend on ρ; it does not prove that a non-zero ρ necessarily degrades performance. No lower-bound analysis, sensitivity test, or monotonicity relation is provided, making the claim that “the bias is inherently harmful” insufficiently supported.

2. Possible confounding with non-IID data heterogeneity.
The reported improvements could stem from alleviating data heterogeneity rather than from the aggregation rule itself. Since non-IID strength is not controlled and there is no oracle baseline (directly computing 1/𝑈∑𝐵𝑢𝐴𝑢), the experiments cannot disentangle aggregation effects from data heterogeneity. To establish a causal link, the authors should fix the data partition and vary only the aggregation operator while measuring the correlation between ρ and performance.

3. Overstated claim that the standard FedLoRA rule is “erroneous.”
The paper repeatedly describes the classical FedLoRA averaging as “erroneous” or “fundamentally flawed.” Yet the analysis only shows algebraic non-equivalence, not that this difference is detrimental under general conditions. Without stronger theoretical or experimental isolation, such a statement appears exaggerated.

4. Limited methodological novelty.
The idea of learning aggregation weights on the server side is far from new. Similar learnable aggregation or re-weighted averaging strategies have been extensively explored in earlier federated optimization, distillation, and low-rank model aggregation works (2018–2021). The proposed approach learning separate coefficients
𝑃,𝑄 for the A and B matrices is essentially a linear re-weighted averaging scheme adapted to the LoRA setting. Hence, the contribution is largely engineering-oriented reuse of an existing idea rather than a conceptual innovation.

5. Suggested clarifications.
Explicitly quantify how ρ affects optimization error or final convergence (e.g., linear vs. sublinear dependence).
Add an oracle baseline that uses the true average 1/𝑈∑𝐵𝑢𝐴𝑢 to decouple non-IID effects. Discuss how this approach fundamentally differs from prior learnable aggregation methods rather than merely showing empirical improvements.

6. Alternative perspective: directly addressing non-IID heterogeneity may be more fundamental.
The paper attributes performance degradation to inaccurate matrix aggregation, but an equally plausible explanation is that client-specific differences in 𝐴𝑢,𝐵𝑢 caused by non-IID data are the real source of the problem. Methods such as FRLoRA (ICLR 2025), which reduce client drift through residual accumulation and subspace alignment, inherently make client updates more consistent thereby indirectly shrinking ρ without modifying the aggregation rule itself. In this sense, tackling heterogeneity at its root might offer a more principled and effective solution than merely adjusting the aggregation formula.

**Questions:**

The motivation and theoretical justification remain unconvincing. The claim that the standard FedLoRA aggregation rule is “fundamentally wrong” is overstated and lacks causal evidence. The proposed learnable-weight scheme is not methodologically novel and revisits well-studied ideas under a new name. Moreover, non-IID heterogeneity is not controlled and could fully explain the observed performance gap. In contrast, approaches like FRLoRA that directly mitigate client drift may address the root cause more effectively. Overall, the work appears to be an engineering refinement rather than a conceptual breakthrough.

---

> ### Author Response · Authors · 2025-11-29
>
> > **W1:** Lack of causal proof that aggregation bias harms performance. The paper defines an error term $\rho = \left\| \bar{B}\,\bar{A}^{-1} - \frac{1}{U} \sum_{u=1}^{U} B_u A_u \right\|$ and incorporates it into a convergence upper bound. However, this analysis merely indicates that model convergence may depend on $\rho$; it does not prove that a non-zero $\rho$ necessarily degrades performance. No lower-bound analysis, sensitivity test, or monotonicity relation is provided, making the claim that “the bias is inherently harmful” insufficiently supported.
>
> The main contribution of our theoretical proof is to propose a relationship between the convergence rate and the error between LoRA matrices and its ideal version. This is not proved in the earlier version of FedLoRA works. We believe that our work will set a potential direction for future works to combine the features of error into the convergence proof.
>
> > **W2:** Possible confounding with non-IID data heterogeneity. The reported improvements could stem from alleviating data heterogeneity rather than from the aggregation rule itself. Since non-IID strength is not controlled and there is no oracle baseline (directly computing $\frac{1}{U}\sum B_u A_u$), the experiments cannot disentangle aggregation effects from data heterogeneity. To establish a causal link, the authors should fix the data partition and vary only the aggregation operator while measuring the correlation between ρ and performance.
>
> We thank the Reviewer for the comment. In this response, we elaborate on the significance of the problem addressed in our work. In the existing literature, the primary focus has been on mitigating the discrepancy between the ideal LoRA parameter average and the aggregated LoRA matrices produced by federated optimization. Moreover, an oracle baseline for FedLoRA already exists, namely SLoRA [R1], which computes the ideal parameter average as $\Bar{W} = \frac{1}{U}\sum^{U}\_{u=1}$.
> Furthermore, we also want to discuss the experimental evaluations in our original manuscript. As stated in Appendix C.5, we set the data heterogeneity level to 0.5 across all experiments. In addition, Section 5.4 evaluates the robustness of the proposed methods under varying degrees of data heterogeneity and compares their performance with baseline approaches.
> [R1] SLoRA: Federated Parameter Efficient Fine-Tuning of Language Models, FL@FM-NeurIPS’23 Oral.
>
> > **W3:** Overstated claim that the standard FedLoRA rule is “erroneous.” The paper repeatedly describes the classical FedLoRA averaging as “erroneous” or “fundamentally flawed.” Yet the analysis only shows algebraic non-equivalence, not that this difference is detrimental under general conditions. Without stronger theoretical or experimental isolation, such a statement appears exaggerated.
>
> We thank the Reviewer for the careful consideration of our work. We would like to clarify that this claim is broadly recognized in the literature on federated finetuning, as evidenced by [R1], [R2], and [R3]. A central challenge highlighted in these studies is the discrepancy between averaging LoRA matrices and the ideal average of the full parameter updates. Addressing this error has been consistently identified as a key open problem in the field.
> - [R1]: Selective Aggregation for Low-Rank Adaptation in Federated Learning, ICLR 2025.
> - [R2]: Improving LoRA in Privacy-preserving Federated Learning, ICLR 2024.
> - [R3]: FLoRA: Federated Fine-Tuning Large Language Models with Heterogeneous Low-Rank Adaptations, NeurIPS 2024.

---

> ### Author Response · Authors · 2025-11-29
>
> > **W4:** Limited methodological novelty. The idea of learning aggregation weights on the server side is far from new. Similar learnable aggregation or re-weighted averaging strategies have been extensively explored in earlier federated optimization, distillation, and low-rank model aggregation works (2018–2021). The proposed approach learning separate coefficients $𝑃,𝑄$ for the $A$ and $B$ matrices is essentially a linear re-weighted averaging scheme adapted to the LoRA setting. Hence, the contribution is largely engineering-oriented reuse of an existing idea rather than a conceptual innovation.
>
> We thank the Reviewer for the comment. We agree that the methodology is not entirely new at the current stage. However, the problem we address remains non-trivial, as demonstrated by several recent studies published in reputable venues [R1], [R2], [R3]. Our method attains state-of-the-art performance while introducing minimal computational overhead and avoiding information loss, as evidenced by the experimental results. In addition, given the conventional structure of our approach, we believe it offers a simple yet effective solution for federated finetuning systems.
> - [R1]: Selective Aggregation for Low-Rank Adaptation in Federated Learning, ICLR 2025.
> - [R2]: Improving LoRA in Privacy-preserving Federated Learning, ICLR 2024.
> - [R3]: FLoRA: Federated Fine-Tuning Large Language Models with Heterogeneous Low-Rank Adaptations, NeurIPS 2024.
>
> > **W5:** Suggested clarifications. Explicitly quantify how ρ affects optimization error or final convergence (e.g., linear vs. sublinear dependence). Add an oracle baseline that uses the true average $\frac{1}{U}\sum B_u A_u$ to decouple non-IID effects. Discuss how this approach fundamentally differs from prior learnable aggregation methods rather than merely showing empirical improvements.
>
> We thank the Reviewer for the constructive comment. According to the Reviewer’s comment, we have conducted more experimental evaluations according to different level of random noise added to the model to simulate the phenomenon of $\rho$ and added to the revised manuscript.
>
> > **W6:** Alternative perspective: directly addressing non-IID heterogeneity may be more fundamental. The paper attributes performance degradation to inaccurate matrix aggregation, but an equally plausible explanation is that client-specific differences in $A_u, B_u$ caused by non-IID data are the real source of the problem. Methods such as FRLoRA (ICLR 2025), which reduce client drift through residual accumulation and subspace alignment, inherently make client updates more consistent thereby indirectly shrinking ρ without modifying the aggregation rule itself. In this sense, tackling heterogeneity at its root might offer a more principled and effective solution than merely adjusting the aggregation formula.
>
> We thank the Reviewer for the insightful comment. We acknowledge that mitigating non-IID heterogeneity is a central challenge in Federated Learning in general and Federated LoRA methods in particular. We would like to clarify, however, that the objective of FRLoRA differs substantively from the scope of our work.
> FRLoRA focuses on a single-round federated finetuning paradigm. Although the method executes this single-round fine-tuning procedure across multiple values of $T$ to explore different learning trajectories. This is because, at the beginning of every round,the train model is reset via the update $\hat{W}^0 + B^0_G A^0_G$, where $\hat{W}^0 = W^0 - B^0_G A^0_G$. Each trajectory is still based on a one-shot update and does not reflect iterative multi-round optimization. After all, the learned trajectories are stored via residuals, and then accumulated via $\widetilde{W}^T = \hat{W}^T + B^0_G A^0_G$.
>
> In contrast, our objective is to design a federated LoRA framework that supports progressive learning over many communication rounds. Our analysis, algorithmic design, and theoretical treatment target this multi-round federated optimization setting rather than the single-round finetuning regime considered in FRLoRA.

---

### Official Review · Reviewer_FWSC · 2025-10-31

**Soundness:** 2
**Presentation:** 3
**Contribution:** 3
**Rating:** 4
**Confidence:** 4

**Summary:**

The paper presents FLoRA-NA, a federated low-rank adaptation method designed to improve update accuracy without increasing communication cost. It estimates aggregated LoRA matrices on the server to reduce the divergence between ideal and practical updates, thereby narrowing the local–global generalization gap. Experiments across language understanding, reasoning, and code tasks show that FLoRA-NA achieves state-of-the-art global performance with strong communication efficiency.

**Strengths:**

1. The paper addresses a well-defined problem in FedLoRA, i.e., the inexact update issue and the resulting local–global generalization gap.
2. Extensive experiments across diverse tasks such as language understanding, reasoning, and code generation demonstrate consistent improvements over strong baselines.
3. The paper provides a clear theoretical analysis.

**Weaknesses:**

1. The introduced matrices P and Q may increase communication costs, but a detailed analysis is missing.
2. The experiments focus mainly on language-related tasks; evaluations on other modalities, e.g., vision-language, would better demonstrate the method’s generality.
3. The paper does not discuss the limitations of the proposed method, which would be valuable for understanding its potential weaknesses and applicability boundaries.

**Questions:**

1. The manuscript ignores important implementation details concerning the optimization of the introduced matrices P and Q. In particular, Algorithm 1 does not show how P and Q are updated (e.g., update rules, gradients, local vs. server updates, or whether they are trained jointly with other parameters). Please clarify these points and update Algorithm 1 (or provide a supplement) to include explicit optimization steps.

2. Are the matrices P and Q layer-wise, or are they shared across multiple layers? Please clarify their scope and how they are applied in the model.

3. If P and Q are layer-wise, the additional communication cost introduced by these matrices should not be ignored. Please discuss how this overhead affects overall efficiency and whether it is accounted for in the experiments.

---

> ### Author Response · Authors · 2025-11-28
>
> > **W1:** The introduced matrices P and Q may increase communication costs, but a detailed analysis is missing.
>
> In our work, we want to note that our method causes no communication cost compare to the vanilla version of FedLoRA (e.g., FedIT [R1]). Specifically, and in Corollary 2, we have mentioned that $P,Q$ are not used for the transmission. After the optimization of $P,Q$, they are used to form an estimated version of $\Bar{A}, \Bar{B}$ for the transmission.
>
> + [R1] Towards Building The Federatedgpt: Federated Instruction Tuning, ICASSP 2024.
>
> > **W3:** The paper does not discuss the limitations of the proposed method, which would be valuable for understanding its potential weaknesses and applicability boundaries.
>
> A primary limitation of our method is that it relies on a single parameter set to modulate the magnitude of client updates through a convex combination. Extending this learnable parameterization to operate at the layer level or even the element level represents a promising direction for future work. In addition, the current formulation does not account for the directional influence of client updates. As a result, gradient conflicts across clients may arise, which can hinder generalization performance. In line with the Reviewer's suggestion, we have added a discussion of these limitations in our description of FLoRA-NA.
>
> > **Q1:** The manuscript ignores important implementation details concerning the optimization of the introduced matrices P and Q. In particular, Algorithm 1 does not show how P and Q are updated (e.g., update rules, gradients, local vs. server updates, or whether they are trained jointly with other parameters). Please clarify these points and update Algorithm 1 (or provide a supplement) to include explicit optimization steps.
>
> We thank the Reviewer for the constructive comment. Specifically, in our work:
> - The coefficients $P$ and $Q$ are optimized on the server by directly minimizing Eq. (6) using the uploaded local LoRA matrices $\{A_u^{(i)}, B_u^{(i)}\}_{u=1}^{U}$.
> - The updates are performed using the AdamW optimizer, as described in Appendix C.5.
> - This optimization is fully decoupled from the client-side training of $A$ and $B$; clients neither update nor store $P$ and $Q$.
> Per the Reviewer’s suggestion, we have revised the manuscript accordingly. In particular, we now include the pseudo-algorithm for optimizing Eq. (6) and provide the corresponding time-complexity analysis of the proposed algorithm.
>
> > **Q2:** Are the matrices P and Q layer-wise, or are they shared across multiple layers? Please clarify their scope and how they are applied in the model.
>
> We apologize for any misunderstanding regarding the proposal of $P$ and $Q$ vectors. As stated in the paper, they are indeed client-wise. Specifically, $P, Q \in \mathbb{R}^{U}$ with $U$ is the number of clients. Consequently, the optimization of $P,Q$ achieve a minimal computation complexity when compare with that of directly optimizing the entire LoRA matrices $A,B$.
>
> > **Q3:** If P and Q are layer-wise, the additional communication cost introduced by these matrices should not be ignored. Please discuss how this overhead affects overall efficiency and whether it is accounted for in the experiments.
>
> We appreciate the reviewer for the comment, because extending the learnable parameter set to operate at the layer-wise is a promising direction for future research. However, we want to note that even in case of layer-wise setup, our method causes no communication cost. Specifically, and in Corollary 2, we have mentioned that $P,Q$ are not used for the transmission. After the optimization of $P,Q$, they are used to form an estimated version of $\widetilde{A}, \widetilde{B}$ for the transmission.

---

### Meta-Review · Area_Chair_TUmW · 2026-01-02

**Summary:**

The paper presents a federated low-rank adaptation method to improve update accuracy without increasing communication cost by estimating aggregated LoRA matrices on the server to reduce the divergence between ideal and practical updates. This work has four reviewers, and three of them are negative and tend to reject this work. The scores of three negative reviewers are 4, 4, and 2. And all three negative reviewers are willing to raise their scores. Hence, this work can not be accepted.

**Reviewer Concerns:**

Concerns which are not addressed by the rebuttal:
1. The motivation and theoretical justification remain unconvincing
2. The proposed learnable-weight scheme is not methodologically novel
3. Overall, the work appears to be an engineering refinement rather than a conceptual breakthrough
4. The core technical idea is not convincing
5. Overclaiming novelty

**Reviewer Scores:**

This work has four reviewers, and three of them are negative and tend to reject this work. The scores of three negative reviewers are 4, 4, and 2.

---

### Decision · Program_Chairs · 2026-01-26

Reject